

# Low Level Cloud and Dynamical Features within the Southern West African Monsoon

Cheikh Dione[1], Fabienne Lohou[1], Marie Lothon[1], Bianca Adler[2], Karmen Babić[2], Norbert Kalthoff[2], Xabier Pedruzo-bagazgoitia[3], Yannick Bezombes[1], and Omar Gabella[1]

[1]Laboratoire d'Aérologie, Université de Toulouse, CNRS, UPS, France
[2]Institute of Meteorology and Climate Research, Karlsruhe Institute of Technology (KIT), Germany
[3]Wageningen University and Research, The Netherlands

**Correspondence:** Cheikh Dione (cheikh.dione@aero.obs-mip.fr)

**Abstract.**

During the Boreal summer, the monsoon season that takes place in West Africa is accompanied by low stratus clouds over land, that stretch from the Guinean coast several hundred kilometers inland. These clouds form during the night and dissipate during the following day. Inherently linked with the diurnal cycle of monsoon flow, those clouds still remain poorly documented

and understood. Moreover, numerical climate and weather models lack fine quantitative documentation of cloud macrophysical characteristics and the dynamical and thermodynamical structures occupying the lowest troposphere. The Dynamics-Aerosol-Chemistry-Cloud Interactions in West Africa (DACCIWA) field experiment, which took place in summer 2016, addresses this knowledge gap. Low level atmospheric dynamics and low-level cloud macrophysical properties are analyzed using in-situ and remote sensing continuous measurements collected from 20 June to 30 July at Savè, Benin, roughly 180 kilometers from the

coast. The macrophysical characteristics of the stratus clouds are deduced from a ceilometer, an infrared cloud camera and cloud radar. Onset times, evolution, dissipation times, base heights and thickness are evaluated. The Data from a UHF (Ultra High Frequency) wind profiler, a microwave radiometer and an energy balance station are used to quantify the occurrence and characteristics of the monsoon flow, the nocturnal low-level jet and the cold air mass inflow propagating northwards from the coast of the Gulf of Guinea. The results show that these dynamical structures are very regularly observed during the entire

41-day documented period. Monsoon flow is observed 100% of the time. The so-called 'maritime inflow' and the nocturnal low level jet are also systematic features in this area. According to monsoon flow conditions, the maritime inflow reaches Savè around 1800-1900 UTC on average: this timing is correlated with the strength of the monsoon flow. This time of arrival is close to the time range of the nocturnal low level jet settlement. As a result, these phenomena are difficult to distinguish at the Savè site. The low level jet occurs every night, except during rain events, and is associated 65% of the time with low stratus

clouds. Stratus cloud form between 2200 UTC and 0600 UTC at an elevation close to the nocturnal low level jet core height. The cloud base height, $310 \pm 30$ m above ground level (a. g. l.) is rather stationary during the night and remains below the jet core height. The cloud top height, at $640 \pm 100$ m a. g. l., is typically found above the jet core. The nocturnal low level jet, low level clouds, monsoon flow and maritime inflow reveal significant day-to-day variability during the summer. Distributions of strength, depth, onset time, break up time, etc. are quantified here.



## 1 Introduction

Clouds are an important factor of uncertainty in climate change studies. The low level clouds (LLCs) that develop during the West African Monsoon (WAM) along the Guinean coast likely contribute to this uncertainty because they modify the Earth's energy budget in a region where the dynamics are driven by strong thermal and moisture gradients and deep convection activity

(Knippertz et al., 2011). However, until recently, very little attention was paid to these clouds. Poorly represented in numerical climate models (Hannak et al., 2017), LLCs form during the night and can extend from the Guinean coast several hundred kilometers inland; they last until midday the following day Schrage and Fink (2012); Schrage et al. (2007); Schuster et al. (2013). These authors emphasize the possible link between LLC formation and persistence and the dynamical features in the region, like the monsoon flow and the nocturnal low level jet (NLLJ). However, very few observations of low clouds and

associated dynamical processes are available, which prevents studies of LLC formation and dissolution, as well as numerical climate and weather models validation. Filling the gap of observations and studying the LLC life cycle were therefore the primary goals of the Dynamics-Aerosol-Chemistry-Cloud Interactions in West Africa (DACCIWA) project (Knippertz et al., 2015) with an aircraft (Flamant et al., 2017) and field campaigns (Kalthoff et al., 2017) performed during summer 2016. At three supersites, Kumasi/ Ghana, Savè/ Benin and Ile-Ife/ Nigeria, ground-based measurements were performed (Fig. 1).

The corresponding datasets are described in Bessardon et al. (2018). The present study takes advantage of the comprehensive dataset acquired at Savè/Benin to provide day-to-day descriptions of the statistical macrophysical characteristics of the LLCs and the dynamical conditions in which they develop.

According to Kalthoff et al. (2017), LLCs form most nights at the three supersites instrumented during the DACCIWA field campaign. Their base are roughly around 300 m above ground level (a.g.l.) when they form, and typically rise up to 800 m

a.g.l. at noon on the following day. Defining LLCs using criteria based on a median cloud base fraction of 100 % at Savé and Kumasi, and a net longwave radiation threshold of -10 W/m2 for Ile-Ife, Kalthoff et al. (2017) noted some differences in the onset times of the LLCs. These authors found that the onset times varied on average among the three sites: 2100 UTC at Ile-Ife, 0000 UTC at Kumasi and 0300 UTC at Savè. Beyond this general description, large variability of the LLC characteristics is observed from one night to the next.

To address this situation, it is important to consider the larger scale context of the WAM and its dynamical features. Such work was the focus, of the previous African Monsoon Multidisciplinary Analysis (AMMA) international project (Redelsperger et al., 2006). The primary dynamical feature affecting West Africa during half of the year is the monsoon flow, which is due to synoptic-scale forcing associated with a strong thermal gradient between the cold tongue over the Gulf of Guinea and dry and warm air inland in the Saharan Heat Low (Lavaysse et al., 2009). In the Southern region, the monsoon flow is overlaid by the

African Easterly Jet (Kalapureddy et al., 2010) at roughly 600 hPa. The monsoon flow exhibits seasonal evolution; its northern limit at the surface, called the Inter-tropical Discontinuity (ITD), moves with the apparent latitudinal position of the Sun. The onset of the monsoon flow marked by a typical northward shift of the ITD and moist convection, manifested by mesoscale convective cystems (MCSs), occurs every year around the end of June (Janicot et al. (2008), Sultan and Janicot (2003)). It corresponds with the start of an active phase of the monsoon moist convection over land to Sahelian regions. In 2016, the



monsoon onset was determined to be $21^{st}$ of June by Knippertz et al. (2017). During the DACCIWA field campaign, which took place from 14 June to 31 July 2016, the ITD was located more than 400 km to the north of Savè (ITD at latitude > 12°N, Savè at 8°N). The ITD mean location in June 2016 is indicated in Fig. 1 and estimated using the 15°C dew point temperature (Buckle, 1996).

5    Knippertz et al. (2017) also defined four synoptic phases of the monsoon using synoptic atmospheric circulations based on the precipitation difference between the coast (south) and the Sudanian-Sahelian zones (north): (i) Phase 1 (1– 21 June): This pre-onset phase was characterized by the modulation of the winds and rainfall in the Guinea coastal area, due to three westward-propagating coherent cyclonic vortices between 4 and 13 °N. The onset itself was associated with a breakdown of the African Easterly Jet, Saharan Heat Low and rainfall due to an extra-tropical trough and cold surge over northern Africa.

10   (ii) Phase 2 (22 June – 20 July): This post-onset phase was characterized by a significant increase in low-level cloudiness and unusually dry conditions above. For example around 12–14 July 2016, Knippertz et al. (2017) highlighted a rare situation of a cyclonic–anticyclonic vortex couplet that crossed South West African (SWA). Unusually dry air was observed in SWA at that time, transported with the anticyclonic vortex which had its centre in the Southern Hemisphere. (iii) Phase 3 (21 – 26 July): A westerly wind regime associated with wet conditions. During this phase, another vortex couplet, located a little further north than the previous one, enhanced westerly moisture transported into SWA. (iv) Phase 3 (27 – 31 July): A phase of monsoon recovery characterized by undisturbed monsoon conditions.

A second very important dynamical feature is the NLLJ, which typically forms over land at the end of the day when daytime convective turbulence has ceased. Blackadar (1957) associated the nocturnal jet with the ceasing of turbulence and predominancy of the Coriolis force, which accelerates the wind towards low pressure. However, due to the low latitude and the low Coriolis force in the DACCIWA region, intertial oscillations might not be applicable. Therefore, the formation of the NLLJ may not be fully explained by this classical theory. Also observed during the AMMA experiment in the Sahelian region (Parker et al. (2005), Lothon et al. (2008), Abdou et al. (2010)), the NLLJ settles almost every night in West Africa. Parker et al. (2005) suggested that when turbulence rapidly diminishes, it is then able to respond to the pressure-gradient force. These authors noted that the previous laboratory experiment of Linden and Simpson (1986), which demonstrated that similar interactions between turbulence and baroclinic flows can occur in the case of a negligible Coriolis force.

Studies by Schrage and Fink  (2012) and Schuster et al. (2013) have suggested that the NLLJ may play an important role in LLC formation because of the cold air advection and turbulent mixing that it generates. Climate models tend to underestimate the strength of the NLLJ (Knippertz et al., 2011; Nam et al., 2012; Hannak et al., 2017), and consequently the advection and turbulent mixing that are associated with its occurrence. The NLLJ contributes differentially to horizontal advection as a function of latitude: transport of moisture occurs toward the north at high latitudes (> 10°N) (Parker et al. (2005), Lothon et al. (2008)). From the point of view of DACCIWA region, transport by the NLLJ may be different. Due to the moisture gradient in SWA, which is characterized by less moisture over the Gulf of Guinea than inland, the NLLJ may actually transport drier air northwards, as revealed by Adler et al. (2018) and Babić et al. (2018).

Another aspect of the DACCIWA region is the sea-breeze circulation that may be superimposed on monsoon flow around coastal regions. This circulation could have an impact on the transport of maritime air inland. Adler et al. (2017) and Deetz et



al. (2018) used COSMO simulations to show that the sea-breeze front, south of which the air is relatively cold compared with the northern more convective boundary layer, reaches several tens of kilometers inland during daytime convective conditions. After 1500 UTC, with the weakening of convection and the associated turbulence, the wind increases and the front propagates further inland. Such a phenomenon was first studied along the Mauritanian Coast and termed "Atlantic Inflow" by Grams et al. (2010). Because this late afternoon propagation of the sea-breeze front occurs in a quite different context than that described by Grams et al. (2010), it is called Gulf of Guinea Maritime Inflow (here after MI) in DACCIWA experiment (Adler et al., 2018) and is one of the processes involved in the LLC formation during DACCIWA field campaign (Adler et al. (2018), Babić et al. (2018)). Depending on the location of the sea-breeze front inland when it starts its late afternoon propagation, between 50 and 150 km must be travelled to reach Savè (Adler et al., 2018). Assuming a mean wind of 6 m s$^{-1}$, this would mean that the MI should reach Savè between 1900 and 2100 UTC.

Adler et al. (2018) and Babić et al. (2018) studied the processes involved in the formation and evolution of LLCs. They both highlighted the important role played by the MI and NLLJ within the monsoon flow diurnal cycle. The present study aims to describe the day-to-day variability and mean characteristics of those structures ; the monsoon flow, the NLLJ, the MI and the LLC during the 20 June - 30 July period, based on the DACCIWA dataset. This investigation is a necessary step forward in ensuring a bette understanding of the LLC life cycle : this work should facilitate future case studies and the model evaluations. The subset of the instrumentation deployed at the Savè supersite and used in this study is described in Section 2. Sections 3 provides the method and the criteria used to detect the monsoon flow. It overviews the day-by-day and mean characteristics of the monsoon flow. Sections 4 and 5 present the NLLJ and MI on one hand and the LLC on the other hand. Section 6 suggests a link between these three phenomena by presenting the mean diurnal cycle. A discussion and conclusion appear in Section 7.

## 2 Experimental data

During the DACCIWA field campaign, several remote sensing and in-situ instruments were jointly deployed at Savè by the Karlsruhe Institute of Technology (KIT) and the Toulouse University Paul Sabatier (UPS). These instruments are all described in detail in Bessardon et al. (2018). The collected dataset has been presented in four published studies: Handwerker et al. (2016), Kohler et al. (2016), Wieser et al. (2016) (for the KIT instrumentation) and Derrien et al. (2016) (for the UPS instrumentation). Furthermore, Kalthoff et al. (2017) give a comprehensive overview of the observations made at the three ground-based supersites of the DACCIWA projet, including Savè.

Below, we describe the instruments used in our statistical analysis, according to the object of study.

### 2.1 Wind profiling of the low troposphere

A UHF wind profiler operated by UPS was devoted to the study of the vertical structure of the atmospheric dynamics in the lower and middle troposphere. This 1274 MHz Doppler radar works with five beams. The three components of the wind are retrieved from these beams. This instrument was previously used for various studies of the planetary boundary layer: turbulence




retrieval (Jacoby-Koaly al., 2002), African Easterly Jet analysis (Kalapureddy et al., 2010), the WAM diurnal cycle (Lothon et al., 2008), the NLLJ (Madougou et al., 2012), and offshore winds in high precipitation events (Said et al., 2016).

The radar ran continuously from 19 June to 30 July 2016 at Savè during the DACCIWA field experiment and provided vertical profiles of the wind every two minutes. We block-averaged the data at 15 minute time intervals for our wind analysis
for consistency with other instruments. For our study, we considered the acquisition mode that corresponded to the highest radial resolution of 75 m, which enabled good documentation of the low troposphere from roughly 150 m up to 3 km in height. The UHF profiler data are used here to characterize the monsoon flow, the NLLJ, and the MI.

## 2.2   Low troposphere temperature profiling

A scanning microwave radiometer (humidity and temperature profiler HATPRO-G4) from the KIT was used to analyze the
temperature profiles. The radiometer measured brightness temperature from which integrated water vapor, liquid water path, temperature profiles, and humidity profiles could be retrieved at a time resolution of 3 minutes. Fifteen-minute average temperature profiles are used here to detect the MI with the 302 K isotherm (Deetz et al., 2018) and the fuzzy logic algorithm described below. A systematic comparison of the radiosounding temperature profiles with the HATPRO temperature profiles (not shown) revealed a systematic cold bias of 0.2 K below 550 m, 0.5 above and below 1000 m, and 2 K above and below
2000 m. This funding is consistent with the accuracies noted by Crewell and Löhnert (2007) (< 1 K below 1000 m). Only microwave radiometer measurements below 550 m are used in this study. The available dataset of this instrument covers the 30 June to 30 July time period.

## 2.3   Observation of surface layer conditions

A 7.77 m scaffold was mounted at Savè by UPS to study energy balance and biogenic fluxes. Ten Hz measurements of
air temperature, specific humidity, and three components of the wind were obtained. The energy flux (latent heat, sensible heat, momentum flux) were calculated for the samples at a time resolution of 30 minutes. Additionally, upward and downward components of shortwave and longwave radiation, air pressure, soil temperature and soil moisture were measured every minute.

This station operated continuously from 13 June to 30 July 2016. In this study, we use the sensible heat flux estimates to characterize the surface layer stability.

## 25   2.4   Cloud monitoring

Three co-located devices deployed during the DACCIWA field experiment are used to monitor low clouds: a ceilometer, cloud radar, and an infrared (IR) camera.

A CHM15k ceilometer, which is a 1064 nm wavelength lidar with a 5–7 kHz pulse rate, was installed by the KIT for the continuous monitoring of cloud base height (CBH). The ceilometer was operated from 3 June to 30 July 2016 at a time
resolution of 1 minute and a vertical resolution of 15 m. Manufactured software automatically provided three estimates of CBH, which allowed us to detect multiple cloud layers. Here, we only use the lowest CBH, which corresponds to the low





clouds under focus. We define 'low-level clouds' as clouds with a base height below 1500 m a. g. l.. Adler et al. (2018) used a lower altitude threshold, 600 m a. g. l., which is well adapted the nocturnal stratus clouds. However, a 1500 m height limit allows us to extend our detection of LLCs to shallow convection in general and detect not only the nocturnal low base stratus deck but also the rising and fractioning cloud base during the morning growing convective boundary layer.

More informations on cloud characteristics was provided by the KIT 35,5 GHz cloud radar (i.e.. cloud top height (CTH) and cloud microphysics (rain, drizzle)). The cloud radar operated continuously from 14 June to 30 July 2016. It pointed vertically most of the time and completed complementary scans every 30 minutes. Here, we use the observations of the vertical profiles for the CTH evaluation. After the despiking process, we averaged the reflectivity profiles of hydrometeors over 5 minutes and applied a threshold of -35 dBz to capture the CTH (Adler et al., 2018). Values below this threshold are considered to be related
to clear air above the cloud. Additional details about the CTH retrieval technique can be found in Babić et al. (2018). This algorithm enables a good estimation of the CTH, particularly when the clouds are uniform, which is true in the case of stratus clouds. However, CTHs are difficult to capture for scattered clouds or rain (e.q., cumulus clouds during the daytime). This fact explains some missing CTH estimates during the daytime in our later analysis.

Finally, UPS installed a MOBOTIX S15 cloud camera that monitored the cloud cover all day and obtained visible and IR
pictures every two minutes. The visible image was a full sky image: the aperture angles for the IR channel were $43° \times 32°$ (which corresponds to 158 m $\times$ 114 m in area at a height of 200 m). In this study, we used 5 megapixels IR images, coded in red, green and blue (RGB) components over 256 colors. Here–after, [R,G,B] denotes the relative contributions of red, green and blue of a given pixel, all defined between 0 and 1 (with accuracy of 1/256). The color of a pixel depends on the emissivity of the corresponding sky area and consequently its brightness temperature (uncalibrated). Typically, a low cloud base is seen
as red and a clear sky is seen as blue. Therefore, a homogeneous low cloud deck will create a homogeneous red color image; a fragmented stratocumulus will render an image with colors ranging from red to blue. This instrument is used here to study the horizontal homogeneity of the cloud deck and to define the onset and breakup times of the stratus deck.

## 3   Monsoon flow analysis

### 3.1   Monsoon flow detection

The monsoon flow can be detected according to the wind direction (Kalapureddy et al., 2010). In this study it is defined as the lower layer with a 135 (SE) – 270 (W) horizontal wind sector. The choice to use a quite large wind direction sector, including some westerly winds, is motivated by weak monsoon flow (below 1 m s$^{-1}$) observed during the daytime for which the direction is ill defined and can be influenced by local effects. Despite this large wind sector, vortex circulations, propagated deep convection and Harmattan flow associated with wind sector between the west and south-east are filtered. The height of
the monsoon flow is the level above which the horizontal wind direction is out of the $135 - 270°$ sector for more than 225 m (i.e., 3 UHF wind profiler gates). Very often, a large wind sheared layer between the monsoon flow and the easterlies (African Easterly Jet) above is observed, which makes it difficult to determine the monsoon flow depth. The strength of the monsoon flow is defined as the mean wind speed within this depth.



### 3.2 Monsoon flow characteristics

Applying the previous criteria to the UHF wind profiler dataset, the temporal evolutions of the monsoon flow depth and strength from 1500 UTC on day D-1 to 1500 UTC on day D are calculated for day D from 20 June to 30 July 2016. The results are presented in Fig. 2. Except for 13, 16 and 17 July, dates for which the UHF wind profiler data are missing, the short periods shown in white in Fig. 2 are for a wind direction not falling within the $135 - 270°$ sector. As discussed in the previous section, the strong variability of the monsoon flow depth observed on some days, like 27-28 June, is associated with a large wind sheared layer and a low wind. However, the intraseasonal variability of the monsoon depth (from a few hundred meters to 4 km) and strength (from almost zero wind to 10 m s$^{-1}$) can often be linked to synoptic conditions. Couvreux et al. (2010) found situations to be true during the AMMA project. Several days in a row, such as 20-23 June and 10-12 July, exhibited consistent monsoon flow characteristics over time with a particularly large monsoon flow depth and strong wind speed. The period 10-12 July is included in the vortex phase (9-16 July). During this period, an unusual cyclonic circulation developed and slowly propagated from eastern Mali to Cape Verde along with an anticyclonic vortex in the west-north-westerly direction along the Guinean coast (Knippertz et al., 2017). The wet westerly regime (21 - 27 July) was characterized by a particularly low monsoon flow strength associated with important oscillations of the monsoon depth due to rain events that shifted back to the coast (Knippertz et al., 2017).

Many convective activities occurred at the Savè site during the campaign, which is typical in SWA (Laing and Fritsch , 1993). These phenomena were either detected by rain measurements (Fig. 2) or density current effects at the surface. Table 1 lists the days for which rain and/or density currents are detected after 1500 UTC; such events are likely to have disturbed or interacted with the formation of LLCs. The impact of density currents at the surface, in terms of temperature, relative humidity, specific humidity, wind speed and direction depends on the intensity of the convective cell and its location. A sudden decrease in temperature and an increase in wind speed are typically observed (Table 2). A shift in wind direction is not a reliable criterion because it depends on the convective cell location with regard to the Savè supersite. As listed in Table 2, density currents are largely associated with dryer air, as already discussed by Schwendike et al. (2010). All of the outflow cases listed in Table 1 are associated with a convective cell identified with the rain radar in the supersite surroundings. All of the rainy and density current cases are excluded (16 days out of 41) from the statistics from which we estimate the primary characteristics of the monsoon flow.

A composite 24-hour evolution of the monsoon flow depth, strength, and direction from 1500 UTC on day D-1 to 1500 UTC the day after during the 20 June to 30 July 2016 period enables a discussion of the diurnal evolution of the monsoon flow characteristics (Fig. 3). The median of the monsoon depth shows a weak diurnal evolution from 1500 m a.g.l. in the middle of the night to 2000 m a.g.l. during convective conditions (Fig. 3a). This finding is consistent with what was observed during the AMMA experiment during the full monsoon season. However, the early monsoon period, when the surface sensible heat flux is still important, exhibited a large diurnal cycle of the monsoon depth that was strongly influenced by the vertical development of the boundary layer (Kalapureddy et al., 2010). Unlike the monsoon flow depth, the strength and direction of the monsoon flow indicate a clear diurnal cycle (Figs. 3b and c). The median strength of the monsoon flow is roughly 3.5 m s$^{-1}$ between





noon and 1700 UTC with a 200° direction. The median strength regularly increases between 1700 and 0100 UTC up to 5.5 m s$^{-1}$ with a simultaeaous slight shift in the wind towards the westerly direction. These same changes are observed in wind surface measurements (Kalthoff et al., 2017).

## 4   Maritime inflow and Nocturnal Low-Level Jet analysis

### 4.1   Detection of the Maritime Inflow and Nocturnal Low-Level Jet

A low-level jet is characterized by a maximum wind speed a few hundred meters above the surface of the Earth and a clear minimum wind speed above. This situation implies significant shear below and above the jet core. Because of the occurrence of the low-level jets in various environments and forcings, different criteria can be applied to define and detect them based on vertical wind profiles. One of the first proposed definitions of the NLLJ based on observations was given by Bonner (1968).

This author defined three types of NLLJs based on three different threshold values for the maximum wind speed (12, 16 and 20 m s$^{-1}$, respectively), and the existence of a minimum wind speed above the maximum, and below 3 km a. g. l.. Stull (1988) and Andreas et al. (2000) defined the NLLJ within the first 1500 m, as having a maximum wind speed at least 2 m s$^{-1}$ faster than the minimum wind above it. Baas et al. (2009) defined NLLJs over the Netherlands as having a maximum wind speed at least 2 m s$^{-1}$ below 500 m a. g. l. and a wind speed 25% faster than the local minimum wind speed above.

The detection of the NLLJ is based, in this study, on the use of dynamical and surface stability criteria: (i) the wind direction in the lowest atmosphere below 1500 m is between the south-east and west-northwest with (ii) a maximum wind speed of at least 5 m s$^{-1}$ below 500 m in height (when the NLLJ settles) and at least 2 m s$^{-1}$ larger than the minimum above and (iii) a surface sensible heat flux lower than 10 W m$^{-2}$. This criterion ensures stable to neutral conditions at the surface. The onset of the NLLJ is defined when these three criteria are satisfied for at least two hours. The breakup time is defined at the point

at which the criteria have not been satisfied for at least 1 hour. The use of the surface sensible heat flux as a diagnostic of the stability may be a limitation to this method because this measurement is very local and may not represent atmospheric stability on large spatial scales.

The DACCIWA project focused on a region to the south of the AMMA study area that is affected by coastal phenomena, as sea breeze. Unfortunately, no measurements provided evidence for MI formation and propagation inland. But based on

simulations conducted by Adler et al. (2017) and Deetz et al. (2018), Adler et al. (2018) hypothesized that the MI penetrated 50 – 130 km inland. When convection and mechanical turbulence vanish at the end of the afternoon, two phenomena occur simultaneously: the monsoon flow increases and the MI, characterized by a higher wind speed than the monsoon flow further north, can propagate further inland, possibly up to Savè. Deetz et al. (2018), using COSMO model simulations for the night 2-3 July, characterized the MI arrival by the 302 K potential temperature observed at a height of 250 m. This criterion was applied

to local measurements at the Savè site in order to detect the arrival of the MI. Additionally, a second method was used in this study based on the combination of both an increase in horizontal wind speed and a decrease in temperature. These changes should be the signature of the MI arrival time. This method based on the fuzzy logic method, was used by Coceal et al. (2018) to detect the sea-breeze front around London, in the southern England. Our method for MI detection at Savè follows three steps:



1/ the rate of changes of temperature ($T$) and horizontal wind speed ($ws$) (termed $r_T$ and $r_{ws}$, respectively) are calculated using 30-minute averaged $T$ and $ws$ in the 200 - 550 m and 150 - 525 m layers, respectively, 2/ fuzzy logic functions for $T$ and $ws$, termed $FLF_T$ and $FLF_{ws}$ respectively, are computed using Eq. 1, and 3/ the mean fuzzy logic function, $FLF_{mean}$, combining both the changes in $T$ and $ws$, is the mean of $FLF_{ws}$ and $FLF_T$.

The fuzzy logic function $FLF_x$ for the variable $x$ can be written as:

$$FLF_x(r_x) = \begin{cases} y_1, & r_x \leq (r_x)_1, \\ y_1 + \frac{y_2 - y_1}{(r_x)_2 - (r_x)_1}(r_x - (r_x)_1), & (r_x)_1 < r_x < (r_x)_2, \\ y_2, & r_x \geq (r_x)_2. \end{cases} \tag{1}$$

where $r_x$ is the rate of change of the variable $x$, $(r_x)_1$ (resp. $(r_x)_2$) is a constant value below (above) which $FLF_x$ is equal to $y_1$ ($y_2$). $r_T$ is multiplied by $-1$ to obtain positive changes for decreasing temperature. As in Coceal et al. (2018), $y_1$ and $y_2$ are set to 0 and 1, respectively and $(r_x)_1$ is set to 0 (i.e., no increase in wind speed or no decrease in temperature). Instead of

using the maximum value of $r_x$ divided by two for $(r_x)_2$ (Coceal et al., 2018), we use the 99-percentile divided by two to avoid outliers. In this study, the mean fuzzy logic function $FLF_{mean}$ is computed using equal weights for horizontal wind speed and temperature: the same threshold of 1 is used to detect combined changes in the dynamic and thermodynamic conditions. Note that the use of the fuzzy logic method is meaningful here if the temperature and wind speed changes are combined. However, to better understand how temperature and wind speed changes distinctly impact $FLF_{mean}$, $FLF_{ws}$, and $FLF_T$

are also discussed. $FLF_{ws}$ would be a full weight affected to wind speed (i.e. considering that MI is only characterized by a dynamical change), and $FLF_T$ would be a full weight affected to temperature in the fuzzy logic method (i.e. considering that MI arrival implies only a thermodynamical change). The MI arrival time can then be determined by noting the first time at which the fuzzy logic functions are equal to 1. There are, in this method, important constants and thresholds to fix, and several sensitivity tests have been performed. The results for individual days can differ but, the average and the conclusions remain the

same.

The use of the MI and NLLJ criteria are illustrated for three very different days in Fig. 4. Each panel presents the 24-hour time-height section of wind as measured by the UHF wind profiler and the three fuzzy logic functions $FLF_{ws}$, $FLF_T$, and $FLF_{mean}$. The NLLJ onset times and the two estimates of the MI arrival time (302 K potential temperature and the fuzzy logic method) at Savè are also indicated.

A brief and weak NLLJ settled in the night of 2-3 July between 2100 and 0300 UTC in a very weak monsoon flow (Fig. 4a). The first simultaneous wind increase and cooling is observed at 2000 UTC ($FLF_{ws}$, $FLF_T$ both reach 1), which indicates a clear MI arrival time. $FLF_{mean}$ values are larger than 0.5 until midnight, which demonstrates that the cooling lasts for at least 4.5 hours. For that day, the 302 K potential temperature criteria is in very good agreement with the fuzzy logic method with a detection of MI arrival time at 2015 UTC. The maximum value of the wind vertical profile reaches the threshold of 5 ms$^{-1}$ at

2100 UTC, which is the onset time of the NLLJ. The second example (Fig. 4b) is for the night of 7-8 July, which is analysed in detail by Babić et al. (2018). The fuzzy logic functions are very similar to those of 2-3 July. The estimated arrival times of the MI are 2000 UTC and 2045 UTC using the fuzzy logic method and the 302 K criteria respectively. Like the night of 2-3 July, the maximum value of the jet reaches the threshold of 5 ms$^{-1}$ at 2100 UTC. The particularity of that day is that the NLLJ





vanishes near the surface after sunrise on 8 July. However, the jet shape of the wind persists above unstable conditions; the wind maximum increases up to 1000 m at 1200 UTC. This example illustrates the necessity of the atmospheric stability criterion for NLLJ breakup time determination. The third and last example, the night of 9-10 July (Fig. 4c), illustrates a NLLJ that has hardly detected because it develops within a strong monsoon flow observed all day long. Six other days presenting a similar strong

monsoon flow are observed during the period between 20 June and 30 July. These days correspond to synoptic anticyclonic atmospheric conditions over the Gulf of Guinea that favor a strong penetration of the monsoon flow inland (Knippertz et al., 2017). In such conditions, the NLLJ onset is not easy to determine. Figure 4c shows that the maximum wind is detected above 500 m before 1930 UTC, which prevents us considering this strong monsoon flow as a NLLJ until that time (criteria (i) of the NLLJ detection). The MI reaches Savè at 1800 UTC according to the fuzzy logic method, 2 hours earlier than the two previous

examples. This result appears in accordance with a strong monsoon flow inhibiting the turbulent mixing and pushing the MI front faster and farther inland. On the other days, the cooling lasts until midnight. The detection of MI arrival time according to the 302 K criteria yields a later MI arrival time (1930 UTC).

   Based on these three examples, one can note that large differences occur that make it difficult to determine solid criteria for the detection of the MI and NLLJ if the same scenario appears every day.

## 4.2   MI and NLLJ characteristics

Characteristics of the MI and NLLJ are deduced from a set of days excluding rainy days and density current cases. For the NLLJ, we use the same days that we used for monsoon flow characteristics. For the MI, the 30 June through 30 July period is used (imposed by radiometer data availability), and 12 days are excluded (listed in Table 1).

   The temporal occurrence of fuzzy logic functions equal to 1 for the 30 June – 30 July period is shown in Fig. 5. One can

note interesting results:

   – A large increase in wind speed ($FLF_{ws} = 1$) is observed at all times during the day; the largest ($\geq 5$) occurs between 1700 and 2100 UTC.

   – As expected, cooling occurs between 1700 and 0000 UTC. Contrary to the wind speed, whose fuzzy logic function reaches 1 but rarely remains at that value for hours at a time, the temperature fuzzy logic function reaches this value many

times during the night. This trend implies continuous cooling (Fig. 4). This result is in accordance with the continuous decrease in temperature within the MI from north to the south discussed by Adler et al. (2018). An abrupt, large change in temperature at the front passage with constant temperature behind would have implied a very different temporal occurrence of $FLF_T = 1$. In addition, the continuous cooling can also be explained by the turbulent mixing under the NLLJ core, which mixes the upper layer with lower layers cooled by radiative and sensible fluxes divergence (Adler et

al., 2018; Babić et al., 2018).

   – The $FLF_{mean}$ with values equal to 1 combines the wind increase and the cooling, both of which occur simultaneously during the entire night; the largest number of cases is observed between 1800 and 2100 UTC.





The MI arrival times are shown in Fig. 6a. Four estimates of the MI arrival times are displayed, one using the 302 K potential temperature criterion and three corresponding to the first time when the three fuzzy logic functions, $FLF_{ws}$, $FLF_T$ and $FLF_{mean}$, attain a value of $1$. The most probable MI arrival time at Savè considering only the wind speed increase is between 1600 and 1800 UTC; it is between 1600 and 2100 UTC when we consider only the cooling. The arrival time

deduced from $FLF_{mean}$, which couples with an equal weight cooling and wind speed increase, exhibited a nearly symmetric distribution centered at 1800 UTC; there are earlier arrival times at 1600 UTC and later ones at 2100 UTC. As noted above, the different tests performed to select the constants and thresholds for the fuzzy logic method yield different MI detection times for each day but quite similar distributions for the period of study. These results suggest that the MI arrival time is difficult to detect with local measurements. An MI arrival time distribution centered at 1800 UTC is an unexpected result because it

seems too early considering the distance to be travelled by the MI after the decrease of the turbulence. However, the MI arrival time as detected by the fuzzy logic function $FLF_{mean}$ is clearly linked to the mean monsoon flow in the afternoon (Fig. 7): the stronger the monsoon flow strength in the afternoon between 1200 and 1500 UTC, the earlier the MI arrival. The two exceptionally early arrivals at 1600 UTC shown in Fig. 7 are associated with unusually strong monsoon flow all day (e.g. the night 9-10 July illustrated in Fig. 4c).

The NLLJ onset and breakup can not be detected prior to 1700 UTC and after 0800 UTC, respectively, because of the stability criterion. The most frequent onset and breakup times of the NLLJ are 1900 UTC and 0700 UTC respectively (Fig. 6b). The MI arrival time detected with $FLF_{mean} = 1$, is also indicated in Fig. 6b. One can note that, with the exception of the two exceptionally early MI arrival times, the NLLJ onset and MI arrival time distributions are quite similar to one another. Table 1 summarizes, for each day of the 20 June through 30 July period, the MI arrival time using $FLF_{mean}$, NLLJ onset and breakup

times, in addition to the rainy and density current cases noted before.

Figure 8 shows a 15-minute distribution of the jet core height, strength and direction for the 20 June through 30 July period. The median jet core lowest height (350 m) is observed when the NLLJ settles. The jet core rises after 2200 UTC to reach 700 m at NLLJ breakup. The most likely strength of the jet varies between 7 and 11 m s$^{-1}$ (Fig. 8b), and the direction shifts slightly from 200° at NLLJ onset to 230° at 0300 UTC (Fig. 8c). If the NLLJ core in Niger and Benin observed during the AMMA

campaign (Lothon et al., 2008) was roughly at the same height, the wind speed of the jet core would have been larger in Niger (i.e. 10 - 20 m s$^{-1}$).

## 5  Low-level cloud statistical analysis

### 5.1  LLC detection

West African LLCs have not been extensively studied in the past, generally due to a lack of observations. However, the system-

atic visual observations by local meteorologists at every meteorological station made over a time period of years constitute a precious dataset. Besides this historical approach, several instruments have been used in the context of international projects to estimate LLC characteristics from a vertical point of view: radiosoundings, cloud radar (Bouniol et al., 2012), and ceilometer



(Schrage and Fink , 2012). Satellite data have also been used to analyze the horizontal variability LLCs. Van Der Linden et al. (2015) and Bouniol et al. (2012), for example, made a climatology of the occurrence of the LLCs in this region.

Here, we use a ceilometer to estimate the cloud base height (CBH) (considering only CBHs below 1500 m a. g. l. ), cloud radar to estimate the cloud top height (CTH) and the IR sky camera to define the start and end of the nocturnal low level stratus

clouds (Section 2.4). With the IR sky camera, we have the advantage of retrieving information about the occurrence of LLCs, also their horizontal structure on a local scale.

To estimate the occurrence of stratus clouds at Savè, we consider the average color $[\overline{R}, \overline{G}, \overline{B}]$ defined by the color vector obtained when averaging all [R,G,B] pixels of the IR camera image, as well as the standard deviation $\sigma_{RGB}$. $\sigma_{RGB}$ quantifies the color variability within each image, which represents the spatial variability of the lowest CBH within the 43° x 32° aperture:

$$\sigma_{RGB} = \frac{(\sigma_R + \sigma_G + \sigma_B)}{3} \qquad (2)$$

where $\sigma_R$, $\sigma_G$, and $\sigma_B$ are the standard deviations of the R, G, B components, respectively within the image.

A large standard deviation implies a large variability of temperature (i.e., a mix of clear sky and clouds) or spatial variability of the CBHs; a small standard deviation corresponds to either an homogeneous stratus deck or a fully clear sky.

Low stratus clouds are said to occur when (i) $\overline{R} > 0.5$, $\overline{R} > G$ and $\overline{R} > B$, (ii) $\sigma_{RGB} <= 0.15$, for more than two hours

without any 'break' longer than 30 minutes. Furthermore, the first time when these criteria are satisfied should occur before 06:00 UTC the following morning in order to detect only nocturnal LLC. The first criterion (i) corresponds to the search of a low base, with an average color close to red. The second criterion (ii) corresponds to the search for a relatively homogeneous cloud deck. The added persistence condition helps with defining a consistent nocturnal stratus cloud life cycle and avoids spurious variability. These criteria allow one to estimate, for each night, the onset and breakup times of LLCs, and therefore

deduce their lifetimes.

Figure 9 shows two typical examples to illustrate this method using information provided by the ceilometer and the cloud radar. The evolution of the average color and the standard deviation during two nights (the nights of 7-8 July and 6-7 July are shown in the (a) and (b) panels, respectively) are shown with the CBH estimated from the ceilometer and the CTH estimated from the cloud radar. Onset and break up times deduced when applying the detection method are also indicated.

For both examples, one can see how the smallest steady standard deviation of the IR image color consistently corresponds more to a steady low cloud base as seen by the ceilometer (see, for example, the data from 8 July from 0300 – 0800 UTC or the data from 7 July from 0600 – 0900 UTC). A larger standard deviation corresponds to more variability of the cloud base (e. g. on 8 July around 1100 UTC or 7 July around 0300 UTC or after 1230 UTC).

For the night of 7-8 July (Fig.9a), the criteria for LLC detection are satisfied from midnight until 1030 UTC. This period

defines the LLC life time this day (table 1). Before midnight, the sky is clear for a large part of the time (blue color with the IR camera); some clouds passed between 1000 and 1500 m a. g. l. The cloud base heights are variable according to the ceilometer; the IR color and $\sigma_{RGB}$ do not satisfy the LLC detection criteria. After 10:30 UTC, the lifting and fractioning of the cloud base with the developing convective boundary layer, when they are large enough, define the end of the stratus LLC (less steady





red-pink color, large $\sigma_{RGB}$ for a long duration of time). We consider the preceding lifting of the stratus deck from 0730 to 1030 UTC as being part of the LLC life time (i. e. they meet our criteria for LLC detection), which is the start of the so-called 'convective phase' according to Babić et al. (2018), who focused on the night of 7-8 July 2016 in their case study.

For the night of 6-7 July, the criteria for stratus LLC is satisfied from slightly after midnight until 1230 UTC (Table 1). The variability of cloud base found between 0200 and 0400 UTC, and revealed by higher $\sigma_{RGB}$ and smaller $\overline{R}$ values, does not persist long enough to interrupt the LLC definition.

Adler et al. (2018) used the ceilometer to estimate nocturnal LLC characteristics (CBH, onset, and break up times). These authors differentiated between LLC stratus deck and preceding stratus-fractus using two lower bound thresholds of 90 % and 50 %, respectively, for the cloud fraction computed within a 60 minutes time interval. An example of those two phases can be seen in the night of 6-7 July (Fig.9b). One can see that low clouds occur between 2100 UTC and midnight; they have a slightly higher base than later in the night, and there are long periods without clouds. This phase corresponds to the 'stratus-fractus' defined by Adler et al. (2018), which precedes the 'stratus phase'. Due to our criteria related to the persistence of LLCs, this phase is excluded from our detection method. We instead detect the 'stratus phase' that corresponds to full deck LLCs. Note that a very short 'stratus-fractus' phase can also be seen in the previous example of the night of 7-8 July (Fig.9a), from 2330 to 0010 UTC, which is also consistent with the work of Adler et al. (2018) and Babić et al. (2018).

Clouds after 1200 UTC for both cases correspond to convective shallow cumulus (within the 'convective phase' noted by Babè et al. (2018)), as revealed by both a larger IR image standard deviation $\sigma_{RGB}$ and the very large scatter in the ceilometer-based CBH between 1000 and 3000 m a.g.l. (Fig.9). The shallow cumulus and fair weather cumulus are not detected by our criteria, and indeed not under scope here.

## 5.2 Macrophysical LLC characteristics

Based on the detection criteria explained above, we noted a significant occurrence of LLC during the DACCIWA campaign at Savè: 65% of the nights exhibit the development of a nocturnal stratus deck. This finding is consistent with the work of Kalthoff et al. (2017).

This prevalent occurrence is revealed by Fig. 10, which shows the evolution —from 15:00 UTC one day D-1 to 18:00 UTC the following day— of the mean color of the sky camera image over the entire study period from 20 June through 30 July 2016. The MI arrival time and onset and breakup times of LLC and the NLLJ are also indicated.

Figure shows some intra-seasonal variability. The pronounced occurrence of LLCs from 26 June through 11 July corresponds to the post-onset phase of the WAM. It is interrupted from 12-17 July by the drier vortex period identified by Knippertz et al. (2017) and noted above. From 23-26 July, the westerly moist regime observed (not shownhere but seen in the UHF wind profiler data of Kalthoff et al., 2017) is associated with more rain (nearly on a nightly basis). This situation prevents LLCs from being detected using the sky camera.

The shortest lifetime of the LLCs was observed on 30 July. On this day the onset LLCs is observed very late. For some cases, the breakup time is observed after noon, as on 3 July. A large variability in LLC characteristics is observed: some nights



exhibit steady LLCs, and other nights have lots of intermittent LLCs(e. g. 11, 21, 29 July). It is clear from Fig. 10 that the LLCs generally form several hours after the onset of the NLLJ (more than 3 hours) and clear up after the NLLJ breakup time.

Figure 11 gives more quantification of those time characteristics. It displays the distribution of the LLC onset time and break up time over the same set of 25 selected days as the monsoon flow statistics discussed above. Large variability in onset and
breakup times of LLCs is indeed observed. Twenty-six percent of LLC onsets are observed before 0000 UTC, and 78% are observed before 0300 UTC. All breakup times are observed after sunrise: 33% of LLC breakup times occurred before 1000 UTC, and 74% occurred before 1200 UTC. We found that LLCs always form more than 3 hours after the onset of the NLLJ and always clear up after the NLLJ breakup time.

## 6   Diurnal cycles of the NLLJ and LLCs

Based on the detection of the NLLJ and LLCs discussed before, we now discuss their composite diurnal cycle.

Figure 12 shows the composite diurnal cycle of the NLLJ core height, LLC base height, and LLC top height computed using the median of those diagnostics over all considered days as indicated previously. The shaded areas correspond to the standard deviation of the ensemble. Note that calculations of the median and standard deviation are possible only when more than 5 cases can be taken into account. Thus, for the NLLJ core height, between 10 and 25 days are taken into account between 1800
(D-1) and 0630 UTC. For the CBH, between 10 and 25 days are taken into account from 00:00 - 11:40 UTC. For the CTH, between 10 and 25 days are taken into account between 04:00 and 11:20 UTC.

Figure 12 provides a schematic summary of the articulation of the NLLJ with the low level stratus that we observe during the DACCIWA field experiment at the Savè site for the 25 days considered in the statistics.

The median NLLJ core height remains lower than 350 m a.g.l. until the onset of the LLC. Once the LLC starts to form,
the NLLJ core rises within the stratus cloud until it dissipates in the morning with increasing instability. LLC of $340 \pm 90$ m thickness form on average at the NLLJ core height. This average thickness is very similar to that found by Adler et al. (2018) based on all days of the Intensive Observation Period ($340 \pm 80$ m, over 15 IOP (Intensive Observation Period) days). The CBH is usually stationary during the night, from 00:00 - 06:00 UTC. It sharply increases after sunrise or later when the convective boundary layer develops.

## 7   Discussion and conclusions

An analysis of a 41-day DACCIWA dataset during the WAM in 2016 in Benin enabled a quantified documentation of low level clouds and dynamical structures in the low troposphere. A key need for numerical weather and climate models before DACCIWA motivated the field campaign.

The months of June and July in West Africa are characterized by deep convection, and the conditions at Savè supersite are
frequently disturbed by MCSs. Among the 41-day period, 16 days have been excluded from the quantified statistics because rain and/or density currents were measured at Savè. However, many other MCSs may have impacted the flow between the coast



and Savè without their being detected at Savè. These MCS are an important part of the monsoon in West Africa and participate in the variability of the low level cloud and dynamical structures discussed below.

The intraseasonal variability of the synoptic situation analyzed by Knippertz et al. (2017) impacts not only the monsoon flow depth and intensity, but all low troposphere structures. For example, a stoppage of the WAM activity in mid-July with an

unusual cyclonic dry air mass circulation weakened the monsoon flow and the NLLJ and inhibited stratus cloud formation at Savè. Besides the day-to-day variability of the monsoon flow due to large scale atmospheric conditions, the median diurnal cycle of the monsoon flow exhibits a 1500-m south-westerly monsoon flow of 6 m s$^{-1}$, which slightly deepens (2000 m), slows down (4 m s$^{-1}$), and turns to the south during the day.

The advection of cool air at Savè is due to two processes already noted by Adler et al. (2018) and Babić et al. (2018): MI and

the NLLJ. The two processes occur because of the late afternoon decrease of the turbulence due to convection. Considering the MI propagation time to reach Savè, both processes are hypothezed to be observed at Savè between 1700 UTC and 2100 UTC. The MI is rather unknown and poorly observed, but its passage at a given location is expected to imply an increase in the wind speed and a decrease in the temperature. The fuzzy logic method applied in this study to detect the MI arrival time leans on these expectations and depends on the weight attributed to the changes in temperature and wind speed. The NLLJ has

been extensively studied in different places around the world but the detection of its time settlement is for sure a question of criteria, particularly the threshold for the wind jet core (5 ms$^{-1}$ in this study). Despite these inherent limitations of criteria-based methods, MI arrival and NLLJ settlement times are estimated at Savè in June and July 2016. The MI arrival time at the Savè site occurs between 1600 and 2100 UTC; the most frequent occurrence is at 1800 UTC. The NLLJ is a systematic feature within the monsoon flow, with a jet core wind speed between 7 and 11 m s$^{-1}$ within the [200°-300°] south-westerly sector at

a height of about 350 m. It usually sets up before 21:00 UTC and ceases before 0700 UTC.

The large time range for MI arrival time at Savè is due to the large range of monsoon flow strength 1-5 m s$^{-1}$ with few cases above 6 m s$^{-1}$. It also depends on the capability of the monsoon flow to balance turbulent mixing, which slows the front progression in land. The MI arrival time is then concomitant with the NLLJ onset; the two processes, as we defined them in this study, are difficult to distinguish. The acceleration of the monsoon flow through the NLLJ over West Africa causes the ITD

to penetrates about 100 kilometers further north every night (Pospichal and Crewell., 2007; Lothon et al., 2008). It is plausible that the MI contributes to this process as well; the Gulf of Guinea/continent contrast and breeze front may come into play as well.

Low level stratus is a persistent phenomenon that occurs 65% of nights. It forms more than 3 hours (average 6 hours) after the NLLJ onset at the jet core height, and it often persists until noon (80% of days). We have shown that LLC forms with their

base at the same height as the NLLJ core; the latter then rises as the cloud deepens. This situation reveals how closely related these processes are:

– The advection of cold air, associated with the monsoon flow, and more abruptly with the MI and the NLLJ;

– The radiative and flux divergence that occur at the end of the day and cool and stabilize the surface layer;



- The turbulent mixing that occurs at the NLLJ core flanks, which can distribute the cooling from the surface up to the cloud layer height.

The relative contribution of those processes is addressed in Adler et al. (2018) and Babić et al. (2018). These authors focused on the formation and evolution of the LLC in Benin during the WAM.

5 *Data availability.* After the DACCIWA embargo period, the data of Savè supersite will be available on the baobab database (Derrien et al., 2016; Handwerker et al., 2016; Kohler et al., 2016; Wieser et al., 2016) for scientists interested in boundary-layer studies in southern West Africa.

*Author contributions.* Cheikh DIONE, Fabienne LOHOU, Marie LOTHON, Norbert KALTHOFF, Bianca ADLER, Yannick BEZOMBES, Omar GABELLA, Xabier PEDRUZO-BAGAZGOITIA participate to the DACCIWA campaign and data processing. Cheikh DIONE prepare
10 the manuscript with the contributions from all co-authors.

*Competing interests.* The authors declare that they have no conflict of interest

*Acknowledgements.* The DACCIWA project has received funding from the European Union Seventh Framework Programme (FP7/2007-2013) under grant agreement no. 603502. The authors thank also Laboratoire d'Aérologie, Université de Toulouse, CNRS, UPS, France and KIT (Karlsruhe Institute of Technology) and UPS (Université Toulouse) for helping to install the equipment as well as the people from
15 INRAB in Savè for allowing the equipment on their ground.



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





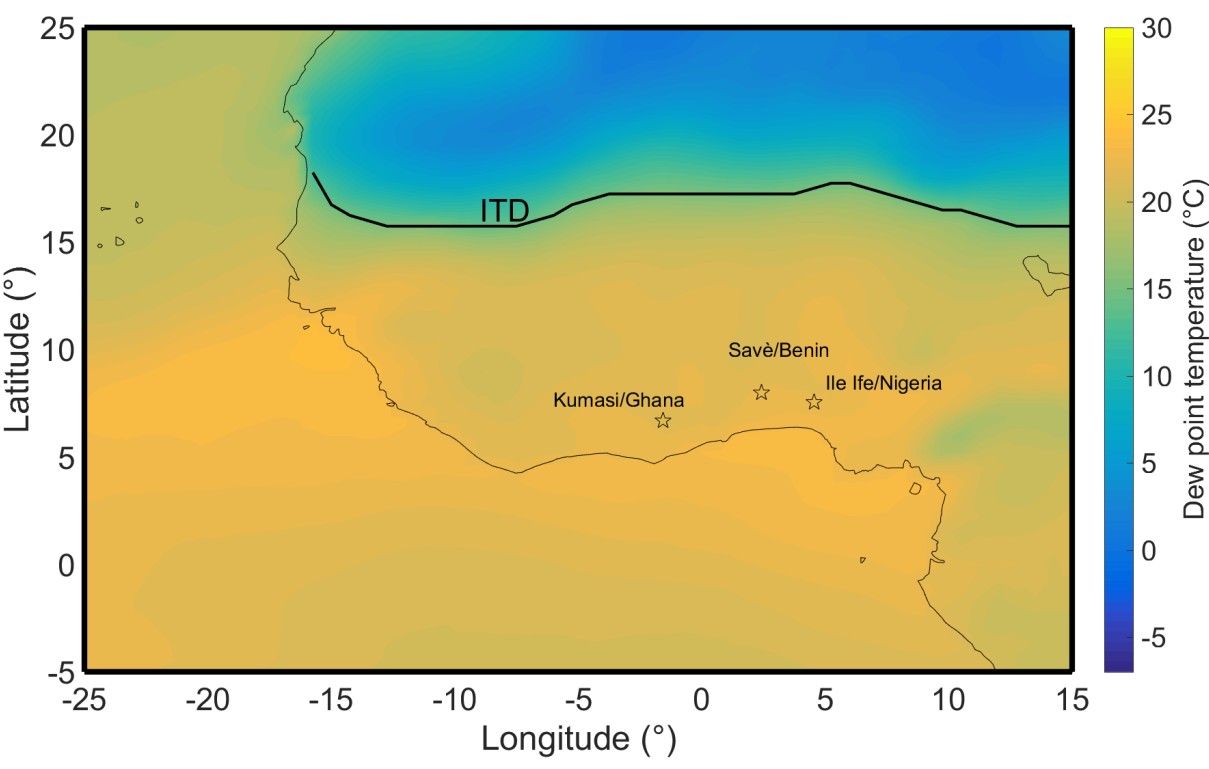

**Figure 1.** The stars indicate the locations of the three instrumented supersites during the DACCIWA field campaign from June-July 2016 in West Africa. The color bar refers to the dew point temperature at a height of 2 m from the ECMWF re-analyses. The intertropical discontinuity (ITD) location (thick blue line) is estimated using the dew point temperature isoline of $15°C$.



**Figure 2.** Diurnal evolution of the monsoon flow (a) thickness and (b) strength at Savè from 1500 UTC on day D-1 to 1500 UTC on day D during the course of the 20 June through 30 July 2016 time period. Vertical black solid lines delimit the different phases of the monsoon described by Knippertz et al. (2017). Vertical dashed lines indicate the vortex phase included in the post-onset phase. White windows indicate missing data or periods for which the monsoon flow was not defined (i.e. a wind direction beyond the 135-270° sector). Grey squares indicate rainy conditions.





**Figure 3.** Temporal evolution of the distribution of the monsoon flow (a) depth, (b) strength and (c) direction at Savè for the period between 20 June and 30 July. The black lines in panels (a), (b) and (c) indicate the median monsoon depth, strength, and direction, respectively.



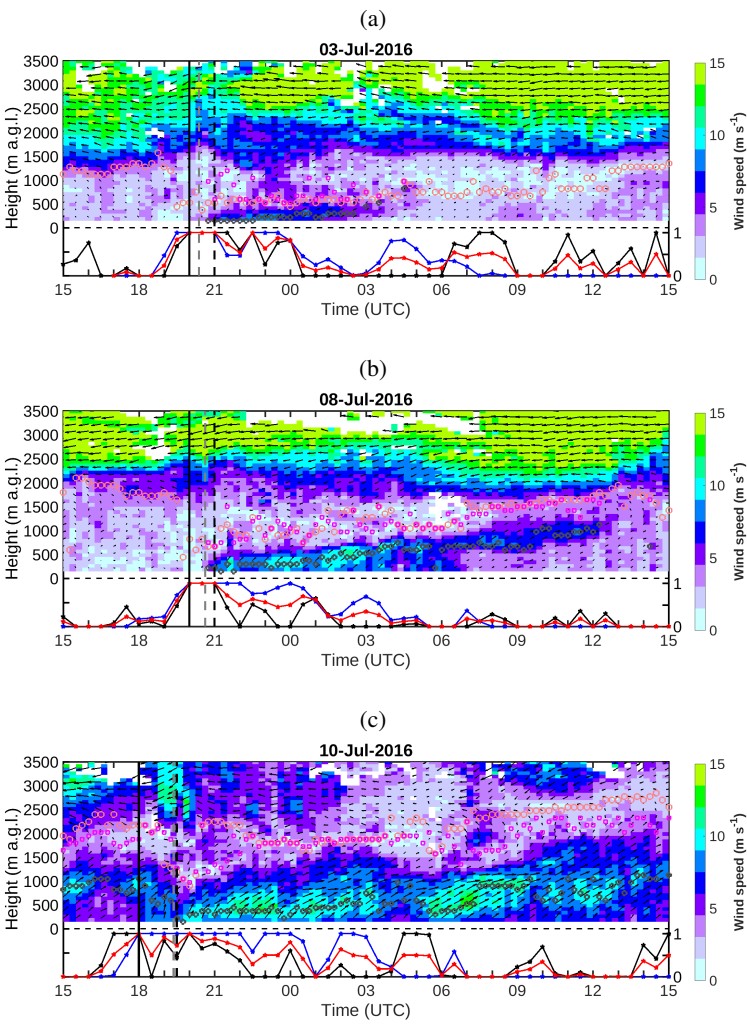

**Figure 4.** Time-height sections of (color) wind speed and (arrows) direction from the UHF wind profiler, on the nights of (a) 2-3 July, (b) 7-8 July, and (c) 9-10 July 2016. The grey open circles indicate the jet core height detected with a maximum wind speed of at least 5 m s$^{-1}$, the magenta rectangles indicate the height of the minimum wind speed above the jet core and pink open circles indicate the monsoon flow depth. The black, blue, and red lines indicate the three fuzzy logic functions of the wind speed, temperature and their mean, respectively.



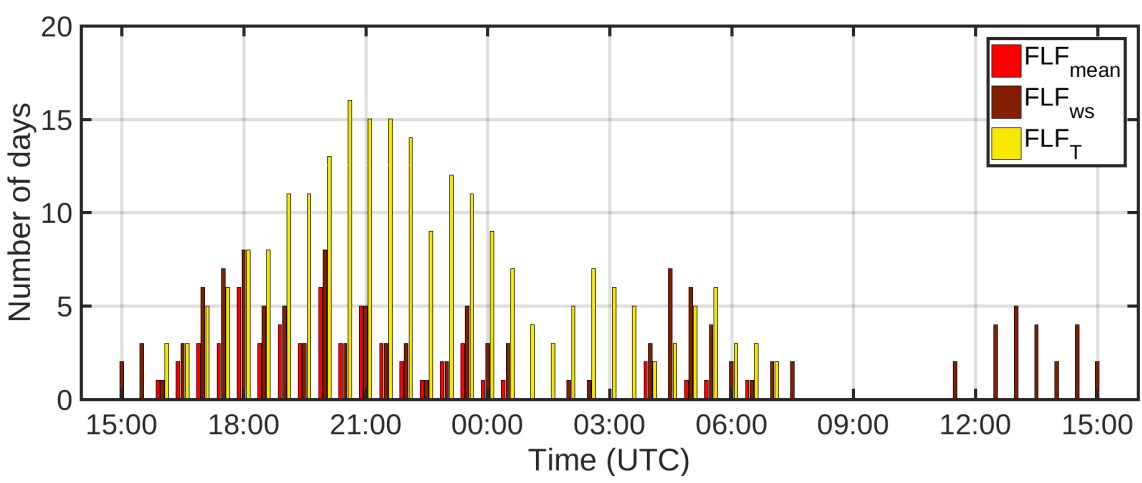

**Figure 5.** Temporal occurrence of fuzzy logic functions equal to 1 for temperature ($FLF_T$), wind speed ($FLF_{ws}$), and mean fuzzy logic function ($FLF_{mean}$) for the period between 1 July and 30 July. period.

**Figure 6.** (a) Distribution of MI arrival times determined using the 302 K potential temperature criterion and the fuzzy logic function method ($FLF_T$, $FLF_{ws}$ and $FLF_{mean}$) for the period between 1 July and 30 July 2016. (b) Distribution of NLLJ onset and breakup times at Savè during the period from 20 June to 30 July 2016. The MI arrival times detected with $FLF_{mean}$, already presented in the upper panel, are also indicated.





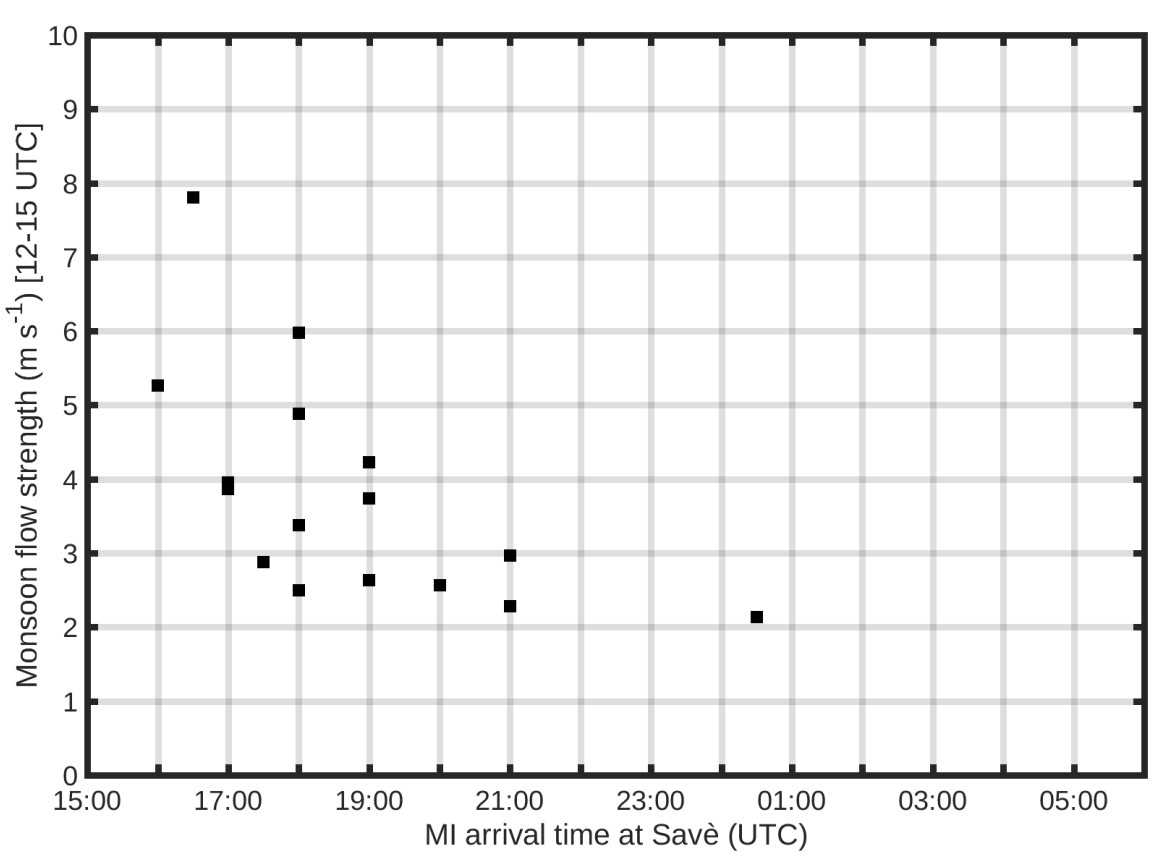

**Figure 7.** Scatter plot of the average monsoon strength between 1200 and 1500 UTC and the MI arrival time using $FLF_{mean}$ =1, for the
time period between 1 July and 30 July 2016.





**Figure 8.** Distribution of the NLLJ (a) core height, (b) strength and (c) direction at the Savè supersite. The color bar indicates the percentage of days (computed using the number of nights with NLLJ onset during the period 20 June through 30 July period). The black lines in panels (a), (b), and (c) indicate the median jet core height, its strength, and its direction, respectively.





**Figure 9.** Time series of the LLC macrophysical characteristics observed on nights (a) 7-8 July 2016 and (b) 6-7 July at Savè. The colored band indicates the time series of the average image obtained every 2 minutes with the IR cloud camera. This time series is plotted arbitrarily at 0 m abscissa for the sake of clarity. The grey line indicates $\sigma_{RGB}$ (with its scale on the right axis) and the dashed grey line indicates for the threshold (0.15) above which the LLC deck is not considered to be homogeneous. Black dots indicate the CBHs of the LLCs estimated from the ceilometer. Blue dots indicate the CTHs of the LLCs estimated from the cloud radar. The vertical black and red dashed lines indicate from left to right the NLLJ and LLC onset and breakup times.




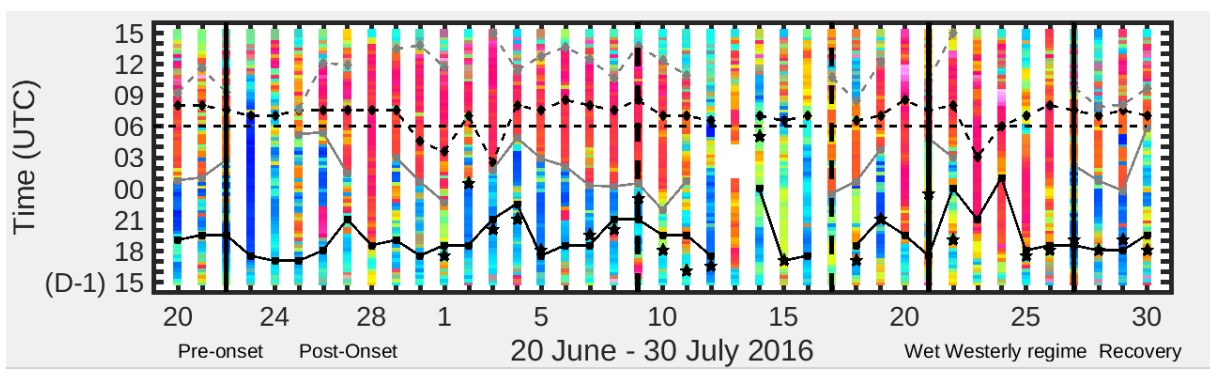

**Figure 10.** The time series of the onset (solid lines) and breakup times (dashed lines) of NLLJ and LLCs are shown in black and gray, respectively. The temporal evolution (from 18:00 UTC the day before to 18:00 UTC the current day) of the mean color of the IR cloud camera image is shown by colored bars. The dashed horizontal black line indicates the sunrise time at Savè. The black stars indicate the arrival time of the MI at Savè using the mean fuzzy logical function.





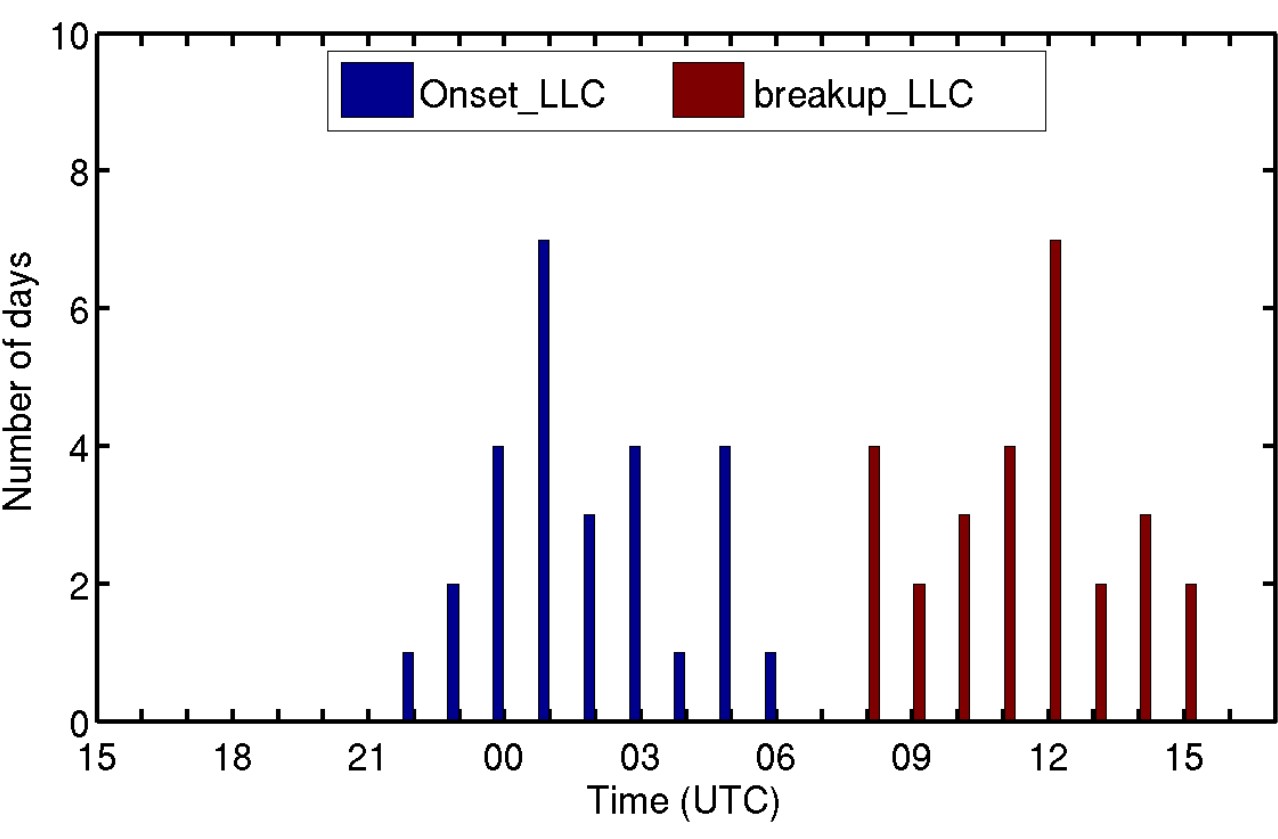

**Figure 11.** Distribution of (blue) onset and (yellow) breakup times of LLCs at Savè from the period 20 June through 30 July 2016.



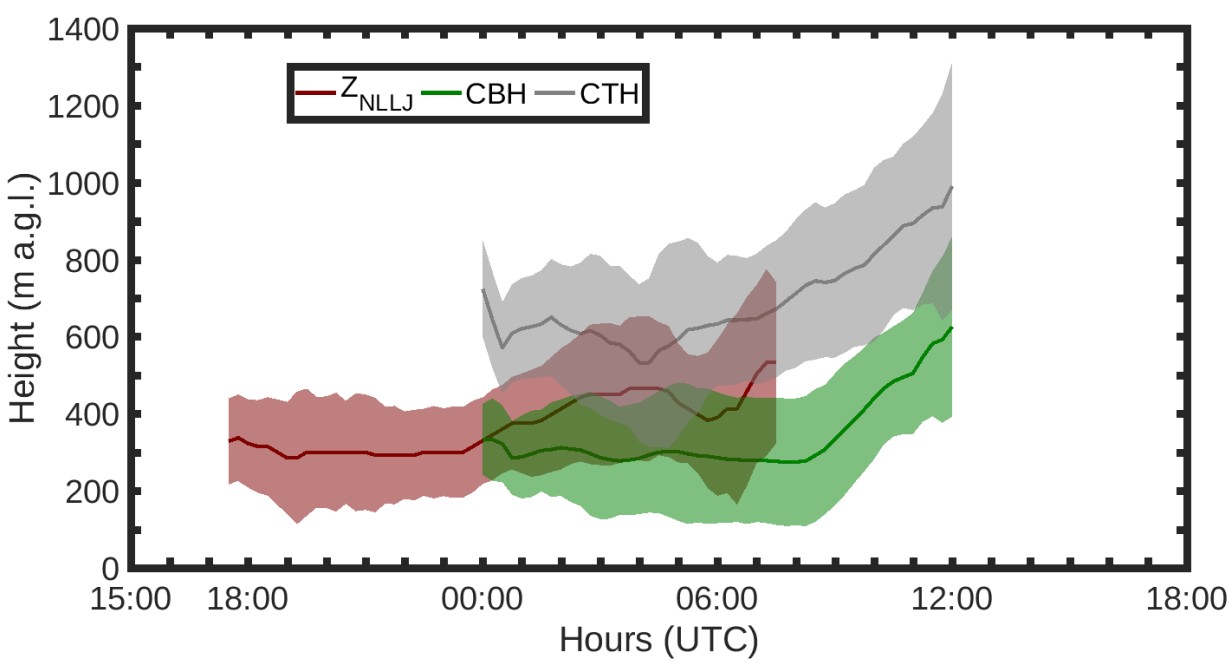

**Figure 12.** Median diurnal cycle of the NLLJ core height (red solid line), cloud base height (green solid line) and cloud top height (gray solid line) from 20 June through 30 July 2016 at the Savè supersite. Shaded areas indicate variability ($\pm\sigma$).



**Table 1.** Characteristics (onset and breakup time) of the NLLJ and LLCs and the arrival time of the MI at Savé during the 20 June – 30 July 2016 period. All times are indicated in UTC. For times related to the LLCs, ' ' indicates that LLCs were not detectable (because of rain or clear sky). For times related to MI or NLLJ, ' ' indicates missing data or undetectable features.

| June-July nights | Rain onset UTC | Total rain mm | DC time (UTC) | MI arrival time (UTC) | NLLJ onset (UTC) | LLC onset (UTC) | NLLJ break time (UTC) | LLC breakup time (UTC) |
|---|---|---|---|---|---|---|---|---|
| 19-20 | 06:40 | 0.1 | | | 19:00 | 00:46 | 08:00 | 09:12 |
| 20-21 | | | 16:31 | | 19:30 | 01:02 | 08:00 | 11:34 |
| 21-22 | | | | | 19:30 | 02:42 | 07:30 | 09:16 |
| 22-23 | | | | | 17:30 | | 07:00 | |
| 23-24 | | | | | 17:00 | | 07:00 | |
| 24-25 | | | | | 17:00 | 05:12 | 07:30 | 07:36 |
| 25-26 | | | | | 18:00 | 05:24 | 07:30 | 12:06 |
| 26-27 | | | | | 21:00 | 01:26 | 07:30 | 11:52 |
| 27-28 | 20:20 | 2.8 | | | 18:30 | | 07:30 | |
| 28-29 | 00:30 | 0.1 | | | 19:00 | 03:00 | 07:30 | 13:26 |
| 29-30 | | | | | 17:30 | 00:38 | 04:30 | 13:46 |
| 30-1 | | | | 17:30 | 18:30 | 22:40 | 03:30 | 11:42 |
| 1-2 | | | | 00:30 | 18:30 | | 07:00 | |
| 2-3 | | | 20:11 | 20:00 | 21:00 | 01:46 | 02:30 | 14:58 |
| 3-4 | | | | 21:00 | 22:30 | 04:52 | 08:00 | 11:22 |
| 4-5 | | | | 18:00 | 17:30 | 02:52 | 07:30 | 12:44 |
| 5-6 | | | 16:05 | | 18:30 | 02:08 | 08:30 | 13:36 |
| 6-7 | | | 17:07 | 19:30 | 18:30 | 00:16 | 08:00 | 12:28 |
| 7-8 | | | | 20:00 | 21:00 | 00:10 | 07:30 | 10:36 |
| 8-9 | 16:20 | 2.6 | 16:15 | 23:00 | 21:00 | 00:28 | 08:30 | 13:44 |
| 9-10 | | | | 18:00 | 19:30 | 21:56 | 07:00 | 12:18 |
| 10-11 | | | | 16:00 | 19:30 | 00:42 | 07:00 | 10:54 |
| 11-12 | | | | 16:30 | 17:30 | | 06:30 | |
| 12-13 | 17:30 | 12.6 | 16:47 | | | | | |
| 13-14 | 04:50 | 0.1 | | 05:00 | 00:00 | | 07:00 | |
| 14-15 | | | | 17:00 | 17:00 | | 06:30 | |
| 15-16 | | | | | 17:30 | | 07:00 | |
| 16-17 | | | | | | 23:30 | | 10:44 |
| 17-18 | | | | 17:00 | 18:30 | 00:38 | 06:30 | 08:26 |
| 18-19 | | | | | 21:00 | 21:00 | 03:42 | 07:00 | 12:08 |
| 19-20 | 00:10 | 15.9 | 17:45 | | 19:30 | | 08:30 | |
| 20-21 | 15:00 | 1.6 | | 23:30 | 17:30 | 04:48 | 07:30 | 10:26 |
| 21-22 | | | | 19:00 | 00:00 | 02:58 | 08:00 | 15:00 |
| 22-23 | 20:10 | 17.1 | | | 21:00 | | 03:00 | |
| 23-24 | 18:20 | 45.9 | 18:50 | | 01:00 | | 06:00 | |
| 24-25 | 19:50 | 1.8 | 16:38 | 17:30 | 18:00 | | 07:00 | |
| 25-26 | 00:00 | 1.4 | | 18:00 | 18:30 | | 08:00 | |
| 26-27 | | | | 19:00 | 18:30 | 02:12 | 07:30 | 09:42 |
| 27-28 | | | | 18:00 | 18:00 | 00:44 | 07:00 | 07:50 |
| 28-29 | | | | 19:00 | 18:00 | 23:46 | 07:30 | 08:02 |
| 29-30 | | | | 18:00 | 19:30 | 05:46 | 07:00 | 09:36 |



**Table 2.** Variations in surface (7.77 m height) temperature ($\Delta T$), relative humidity ($\Delta RH$), specific humidity ($\Delta q$), wind speed and wind direction outflows that can disturb the MI arrival time and NLLJ onset and participate in the cooling of the atmosphere detected after 1500 UTC. For the wind changes, the wind speed and wind direction averaged within the layer [150–525 m] before and after the event are considered.

| Days | Hours (UTC) | $\Delta T$ (°C) at 7.7 m a.g.l. | $\Delta RH$ (%) at 7.7 m a.g.l. | $\Delta q$ (g kg$^{-1}$) at 7.7 m a.g.l. | wind speed shift (m s$^{-1}$) 150–525 m a.g.l. | wind direction shift (°) 150–525 m a.g.l. |
|---|---|---|---|---|---|---|
| 20 June | 16:31 | -3.1 | 10.2 | -0.90 | 8.0–15.3 | 178–196 |
| 2 July | 20:11 | -2.8 | 18 | 0.92 | 1.6–2.4 | 243–238 |
| 5 July | 16:05 | -2.3 | 6.7 | -0.33 | 3.5–4.5 | 237–278 |
| 6 July | 17:07 | -1.4 | 8.9 | 1.01 | 3.1–4.2 | 176–174 |
| 8 July | 16:15 | -5.8 | 25.6 | 0.18 | 0.7–12.0 | 14–179 |
| 12 July | 16:47 | -7.1 | 32.0 | 0.43 | 4.8–11.4 | 229–300 |
| 23 July | 18:50 | -2.3 | 10.2 | -0.37 | 2.9–7.4 | 190–318 |
| 24 July | 16:38 | -4.2 | 27.5 | 2.46 | 1.9–4.8 | 321–335 |