# Peer review of "Low Level Cloud and Dynamical Features within the Southern West African Monsoon"

_Atmospheric Chemistry and Physics, 2018_

## Referee Comment (RC1) · Anonymous Referee #1 · 5 Jan 2019

**Review of Low Level Cloud and Dynamical Features within the Southern West African Monsoon by Cheikh Dione et al.**

**Summary of manuscript:**

The paper analyzes new observational data from a ground site at Savè, Benin, which was established as part of the DACCIWA campaign. The data analysis focuses on quantifying the diurnal cycle and intra-seasonal variability of factors related to the southern West African Monsoon, associated dynamical features like the maritime inflow (MI) and nocturnal low level jet (NLLJ), and the formation and breakup of low level stratiform clouds.

Monsoon flow was found to occur at some point on every day studied, with the strongest flow occurring at night. Onset of both the MI and NLLJ occurred most frequently between 1600-2100 UTC and breakup of the NLLJ occurred most frequently around sunrise. The distribution of MI arrival times was shifted earlier than expected considering the distance travelled, but strong monsoon flows may explain this result. The paper highlights the difficulty of cleanly separating MI and NLLJ phenomena in observations.

Low level stratiform clouds formed on 65% of the nights studied and usually broke up by 1200 UTC the following day as the planetary boundary layer became more turbulent and deepened. Cloud bases were typically formed near the core of the NLLJ.

The manuscript is well-organized but lacks sufficient detail and clarity in discussing its methods and reporting its results. The data analysis itself seems to be on solid footing but the manuscript requires a substantial amount of editing to provide a clearer accounting of the analysis and its significance.

In addition, a significant amount of further proofreading and editing is necessary for missing units, unlabeled elements on figures and tables, figure legibility, citation format, general typos, and grammar.

**Recommendation:** I recommend acceptance following adequate revision of the manuscript for clarity and completeness in reporting its methods and results.

**Major issues:**

1. It would be more accurate in your title and throughout the paper to refer to "low level stratiform clouds" or something similar rather than "low level clouds." The paper explicitly limits its analysis to stratiform boundary layer clouds and does not dwell on the shallow cumulus clouds that form after the stratus breaks up during the day. These low-level cumuliform clouds also may be worthy of future study for radiative and other implications.

2. Page 1, Line 15: The statement "Monsoon flow is observed 100% of the time" seems to contradict Figure 1 and the definitions in the paper. Did you mean that monsoon flow is observed at some point on 100% of days studied? Such a statement would be supported by the data provided.

3. Figures and tables, generally: There are major issues with clarity on several figures and tables.

a) Figure 2: The gray markers are extremely difficult to distinguish against the background shading. A more contrasting color, such as gold, could make the figure more legible.

b) Figure 4: The vertical lines are not labelled in the figure, described in the caption, or mentioned in the manuscript. My impression is that the solid black line is for the $FLF_{mean}$ MI threshold, the grey dotted line for the potential temperature MI threshold, and the black dashed line for the NLLJ threshold, but this should be made explicit within the figure or in the caption. In addition, there is no indication of what the horizontal dashed line signifies, although it appears to be the zero marker on the Height axis and also a separator between the z-t plots and the FLF plots. As before, the grey markers are barely legible against the background shading.

c) Figure 10: As before, the grey color is incredibly hard to discern. Here it's possible that thickening the lines would be sufficient, although choosing a different color that offered more contrast would work as well.

d) Table 1: It would be helpful to specify in the caption that the $FLF_{mean}$ criterion was used in the MI onset column. In addition, "DC" is never defined in the caption or the text, although I'm assuming it stands for "density current."

**Specific comments:**

1. Title: It would be helpful to add "Observed at Savè, Benin" at the end of the title to better describe the paper. It's not clear from the present title whether the study will focus on model results, satellite observations, site-specific observations, etc., and the most significant portion of the paper is the description of novel observations taken at the site.

2. Page 1, Line 5: What does the term "quantitative documentation" mean in this context? Is it that the clouds are not well-simulated, or that not enough has been published about the simulated cloud properties?

3. Page 1, Line 16: It's not clear what "According to monsoon flow conditions" means in this context. You mention the correlation with monsoon flow strength in the next line, which seems to make this phrase redundant.

4. Page 1, Line 23: Perhaps "and intra-seasonal" should be added in between "day-to-day" and "variability" given the importance of the different monsoon phases and synoptic setups (e.g., vortex circulations).

5. Page 2, Line 13: It's confusing to distinguish between "aircraft" and "field" campaigns — aircraft campaigns are generally considered a subset of field campaign. For example, a NASA data archive defines an atmospheric field campaign as "an observational study planned for a specific location and a defined time period during which measurements are conducted from airborne platforms and/or ground sites to study physical and chemical processes in the atmosphere" (https://eosweb.larc.nasa.gov/field-campaigns). "Ground-based" may be a more appropriate phrase for the supersite data.

6. Page 2, Line 13 (and throughout): The citation year for the Kalthoff et al. paper should be 2018 instead of 2017 to refer to the published version.

7. Page 2, Line 16: Why are only the data from Savè used? It would be helpful to more fully motivate the decision to focus on this site in particular when two others are theoretically available as well.

8. Page 4, Line 1: Please define COSMO before using the acronym.

9. Page 4, Line 18: "On the one hand… on the other hand" generally signifies that two things will be contrasted, but that is not really the case in these sections. A re-write to "Section 4 presents results for the NLLJ and MI and Section 5 for the LLC" or something similar would be better.

10. Page 5, Line 14: It is not clear what "above and below" 1000/2000 m means in this sentence. Is it 0.5 K between 550-1000 m and 2 K between 1000-2000 m? Or something similar? Please clarify.

11. Page 6, Line 29: The vortex circulations, deep convection, and Harmattan flow are filtered out, or excluded, from the analysis, correct? Just saying "filtered" is ambiguous about whether these observed values are excluded or somehow corrected.

12. Page 7, Line 9: The phrase "found situations to be true" is missing some critical information. What did the simulations find to be true? In context it seems that the Couvreaux et al. paper is cited to support the previous assertion about linking synoptic setups to monsoon variability. Perhaps it would be better to just cite the paper at the end of that sentence if you're not making any further points about the study?

13. Page 7, Line 34: It's a bit of stretch to say that Figure 3c indicates a "clear diurnal cycle" in wind direction. Can you in some way quantify that there's a statistically meaningful diurnal difference? It seems likely to me the difference is real, but it's not self-evidently true.

14. Page 8, Line 18: It would be helpful to put "This last criterion" or something similar here to make clear it's only the third criterion that "ensures stable to neutral conditions at the surface."

15. Page 8, Line 30: When introducing the fuzzy logic method, it would be helpful to motivate why this method is necessary/helpful. From the rest of the paper it seems like the 302 K potential temperature threshold works just as well, so the main benefit is the ability to look at wind and temperature components separately?

16. Page 9, Line 11: It would be helpful to rewrite Equation (1) here plugging in the values for $y_1$, $y_2$, and $(r_x)_1$:

$$FLF_x(r_x) = \begin{cases} 0, \ r_x \leq 0 \\ \dfrac{r_x}{(r_x)_2}, 0 < \ r_x < (r_x)_2 \\ 1, r_x \geq \ (r_x)_2 \end{cases}$$

In addition, for transparency/reproducibility, you should provide the numeric values used for $(r_x)_2$ for both temperature and wind speed.

17. Page 9, Line 11: The definition for the mean FLF function should specify whether you're averaging the two other FLF functions (my impression) or taking $r_x$ as the average of the wind speed and negative temperature tendencies.

18. Page 9, Line 13: I can't tell what this sentence about the fuzzy logic method being "meaningful" is actually saying. Meaningful in what sense? Is there some evidence that you want to highlight about this being a meaningful metric?

19. Page 10, Line 8: Is this supposed to be criteria ii)? Also, as written on Page 8, criterion ii) does not make clear the maximum wind speed must be below 500 m, just that the maximum wind speed below 500 m must be at least 5 m/s. You should clarify this criterion.

20. Page 10, Line 14: It is not clear what "if the same scenario appears every day" means here. If every day had the same scenario, it seems like it would be quite easy to determine solid criteria. This sentence should either be written to more clearly state its point or be deleted.

21. Page 10, Line 21: There is a notable period of muted wind speed increases in the morning between 0600-1200 UTC. This seems to contradict the "all times of day" phrasing. In addition, there are also spikes above 5 just before 6 UTC that complicates a simple 1700-0000 callout.

22. Page 11, Lines 3 and 23: The phrases "most probable" and "most likely" suggest some kind of statistical analysis, although none is carried out, or at least documented. If these are conclusions just from visual inspection of Figures 6 and 8, it would be better to say something more along the lines of "most observations fell between the values of…". If you have some threshold (interquartile range? two standard deviations?) being used to define "most probable" or "most likely," it should be reported.

23. Page 11, Line 24: It's not clear how you reach the conclusion that the NLLJ cores from AMMA would have higher wind speeds if they were the same height as those observed in this paper, or what the implications of this are.

24. Page 12, Line 14: Should G and B also be the average values in criterion i)? If not, what fraction of pixels must satisfy criterion i) for the scene to be considered cloudy?

25. Page 13, Line 30: I can't find where earlier in the paper it's mentioned that LLCs cannot be determined during rain events. From Figure 10 it appears that the IR camera continued to collect valid data. Please clarify either here or in an earlier section.

26. Page 14, Line 17: What does "articulation" mean in this context? It's also unclear what exactly is "considered in the statistics."

27. Page 14, Line 27: Did you mean to say "a key need for observations to compare with numerical weather and climate models" or something to that effect? The sentence is missing something as currently written.

28. Page 15, Line 1: It would be helpful to discuss a bit how the MCSs impact the variability results in the paper versus simply asserting they're important, given this was not made much of a focus previously in the manuscript.

29. Page 15, Line 8: Does the flow really turn to the south? From Figure 3, it looks like the flow becomes more southerly if anything, meaning the winds are turning to the north. Saying the winds become more southerly would be clearer.

30. Page 15, Line 29: It would be better to say that low level stratus clouds persist until noon on "80% of days with nocturnal stratus formation" or something similar. Otherwise it looks like this was observed on 80% of all days, which is problematic given that only 65% of days had nocturnal stratus cloud formation to begin with.

31. Page 16, Line 4: This seems very abrupt and incomplete for a conclusion to the paper. The paper would be greatly improved with a final paragraph explaining the broader significance of this work and perhaps suggestions for future directions or uses for the data.

32. Page 16, Line 5: You should state explicitly how long the DACCIWA embargo period will be. It would also be useful to provide a DOI or URL to the baobab database if available.

**Technical corrections:** There are numerous issues of copy-editing (grammar, reference format, etc.) that need further review. Because of the importance of the abstract, I list all the issues that I identified here. I leave the remaining, similar errors to the authors to address in further proofreading unless the mistake impedes understanding or is in an important location (e.g., subhead).

1. Page 1, Line 2: "Boreal" should not be capitalized.

2. Page 1, Line 3: There should not be a comma after "land."

3. Page 1, Line 4: "These" should be used instead of "those."

4. Page 1, Line 9: "Continuous measurements collected" should be changed to "measurements continuously collected" or the "continuous" should be moved to before "in-situ" in the line above.

5. Page 1, Line 11: "Data" should not be capitalized.

6. Page 1, Line 20: "Stratus cloud" should be pluralized.

7. Page 3, Line 15: "Phase 3" should instead be "Phase 4."

8. Page 4, Line 28 & Page 5, Line 8: "Low troposphere" should be "lower troposphere."

9. Page 5, Line 14: There is a missing unit of "K" after 0.5.

10. Page 13, Line 27: Figure 10?

---

## Referee Comment (RC2) · Anonymous Referee #2 · 8 Jan 2019

Review of the study "Low level cloud and dynamical features within the Southern West African Monsoon" by Dione et al.

General comment: This study aims at analyzing the dynamic and the variability of the nocturnal low level jet, the maritime inflow and their connections to the low level clouds thanks to high resolution wind profiler, observations and cloud monitoring. This study is correctly written and the results are clear and well presented. The only main comment is on the main objectives and the contribution of this study with respect to the DACCIWA project and the others studies/papers that should be better highlighted. I recommend minor revisions with the list of detailed comments below.

Abstract: I recommend to the authors to improve the abstract by removing some descriptions and better identifying and emphasing the main objectives and results of this

[Figure]

specific study. The main contributions of this study to the DACCIWA project should be clearly mentioned. This is also true for the introduction.

P2 l33, P3 l23, P3 l31 . . . Nested brackets.

Fig 1: Please clarify that the colors show the mean dew point temperature for all the period or only during the DACCIWA campaign. Also why the ITD is only displayed for June (P3l3).

P5 l19 I am not used to using terms like "Ten Hz . . .", does it mean a high frequency measurement?

P6 l2: The justification of the increase of the cloud base definition is not clear. The reason mentioned is to also detect shallow convection during the morning. But this is not indicated in the objective of the study. Please clarify.

P7 l3 and Figure 2: this sentence is not clear. To avoid confusion, I recommend to change the color when the data are missing.

P7 l27 and Fig. 3: Because there is a strong seasonal cycle during this period (according to Fig 2, it seems there is a thickening of the monsoon layer from the beginning of July, around the 10th), what is the results of a composite study when splitting before and after the 10th of July?

Figure 4: Please clarify what the vertical lines represent.

P11 l2: How do the authors explain the delay in between the FLF derived from the wind and the temperature?

Figure 6a: It is difficult to distinguish the black and brown bars, please change the colors.

P11 l9: Why not using the new ERA5 reanalysis with hourly resolution to detect and analyze briefly the large scale detection of the MI and to better understand these difficulties at local scale?

P11 l13 and Fig. 7: Could you add and discuss the impact of the dates by adding colors for each dot? The monsoon strength and the MI arrival time should be related to the seasonal cycle. Isn't it?

P11 l21 and Fig. 8: same comment as previously. Is there any difference between June and end of July?

P11 l21: The authors do not discuss the intensity of the wind that is maximum at 2.00 am. I expected later as shown by Ruchith and Raj (2015). Ruchith, R. D., & Raj, P. E. (2015). Features of nocturnal low level jet (NLLJ) observed over a tropical Indian station using high resolution Doppler wind lidar. Journal of Atmospheric and Solar-Terrestrial Physics, 123, 113-123..

P13 paragraph starting l27: The authors suggest the impacts of the difference phases of the monsoon. Why this is not taken into account this when calculating the composite studies?

P14 l29: The authors mention other sites in introduction but it seems they are not used in that study. So it is not necessary to mention them and to put them on the map Fig. 1.

---

## Referee Comment (RC3) · Anonymous Referee #3 · 21 Jan 2019

Review of "Low-level cloud and dynamical features within the southern West African monsoon" by Cheikh Dione et al.

Review summary

The authors describe the occurrence statistics of the nocturnal low-level jet, the maritime inflow, and stratus deck during a 40-day observation period of the DACCIWA campaign at Savè, Benin. The jet and inflow are identified from UHF wind profiler measurements and radiometer-derived temperature profiles. The stratus deck is identified using IR "RGB" measurements. The authors identified the dynamical features on 20-25 days out of 40 (when there was no appreciable precipitation or density current) and the stratus deck on most of those days. Comparing the onset and breakup, the stratus deck was found to initiate approximately 3 hours after the onset of the maritime

inflow, while the breakup of the jet was found to occur around sunrise and the breakup of the stratus after sunrise.

The manuscript is generally well-written with a clear structure. The figures present some nice and valuable results, although the figures are sometimes too busy to easily interpret. Some of the statistical findings are reported using vague or incorrect terminology. Overall, the scientific concerns are minor and this paper could be accepted after minor revisions.

Major comments

1. Conclusion / novelty

The authors lean a lot on the work presented in Adler et al. (2018) and Babic et al. (2018). When reading the conclusions, the emphasis appears to be on the findings of those two papers, e.g. the list starting at line 31, page 15. Instead, the authors should highlight in their conclusion how their Figure 12 synthesizes their results. For instance, one could identify three key periods, e.g. 1800-0000, 0000-0700, and 0700-1200 (more or less as done in lines 10-24, page 14). This Figure 12 provides a broader context for the case-study type and process analysis done in the previous studies. The main point here is that from the conclusions, it is unclear what the novelty is of this particular paper (although the introduction does provide this in line 13-19 on page 4).

2. Busy figures

The amount of information condensed into single figure panels is impressive, but it makes it very difficult to interpret some of these. One might imagine using these figures in a presentation and certain features will be difficult to highlight. Specific issues are:

Figure 2. The grey squares are difficult to see. How important are these for this figure? The rainy conditions could be presented in separate panels, although that would shrink Figure 2. Alternatively, the authors could present the rainfall information in a separate Figure using the same day-hour axes. If the information is not crucial to the paper, it

could be provided as a supplementary figure. As it stands, the information is getting lost.

Figure 4. The three markers are difficult to discern in this figure. A solution could be to (1) remove the wind barbs to a separate figure and (2) replace the open markers with slightly larger, filled, black markers of different shapes. As a separate point, the barbs are not intuitive to interpret, as they are shown against a height axis. If the barbs were placed in a separate panel, the authors could also colour code them or use a filled contour plot to emphasize different cardinal or intercardinal directions.

Figure 10. Although the figure is visually fun, it is difficult to read. The preceding analysis means that the colours are no longer necessary. Without the colours, it would be much easier to interpret the relationship between the onset and breakup of the jets and clouds.

Minor comments

Page 3, line 4. Please mention the source of "dew point temperature" used in this paper.

Page 3, line 18. Please rephrase or clarify in the text what is meant by "convective turbulence".

Page 3, line 20. What are "intertial oscillations"?

Page 3, line 23. What does "it" refer to? The NLLJ?

Page 5, line 6. Please give the exact limits of the profiler data, rather than "roughly 150 m".

Page 5, line 14. "above and below" this phrasing does not make sense. It suggests that the bias is both 0.5 and 2.0 K between 1000m and 2000m.

Page 5, line 15. "funding" should be "finding".

Page 5, line 19-24. Given that the UHF data are block-averaged to 15 minutes, and given that it is interpreted alongside the sensible heat flux. Shouldn't the latter also be block-averaged to 15 minutes? Please specify the averaging performed on these data.

Page 5, line 30. "manufactured" should be "manufacturer"

Page 6, line 7. "most of the time" and "complementary scans". Please be specific. Did the radar perform a volume scan every 30 minutes? How long did the volume scan take, e.g. 5 minutes? Does that mean that you have five 5-minute estimates of cloud-top height per 30-minute period?

Page 6, line 14-22. This analysis seems really nice and original. Is it designed in this study? Perhaps the authors could emphasize this more. If not, please provide references.

Page 6, line 29. Should "height" be "top" (of the monsoon flow).

Page 6, line 33. "depth" should be "layer".

Page 7, line 29. The monsoon depth is less than 1500m in the middle of the night.

Page 7, line 31-33. The authors use a reference from 2010 to describe the status of the monsoon in their 2016. Please consider rephrasing this sentence.

Page 8, line 18. It is important here to note the temporal resolution of surface sensible heat flux, if it is different to the other measurements (see previous comment for page 5).

Page 8, line 29. How is the 302 K potential temperature measured? Is it based on the radiometer profiler? Please specify.

Page 9, line 15-16. "affected" should be "applied".

Page 9, line 18-20. Please enlighten the reader to what range of thresholds are appropriate for r-ws and r-T, and which values were chosen for the subsequent analysis.
Page 10, line 1-2. "the wind maximum increases" – it is the "height" of the wind maximum that increases. Please rephrase.

Page 10, line 14. "if the same scenario appears every day" – surely, the authors mean that it is difficult to determine criteria if "different" scenarios appear each day? (i.e. the opposite)

Page 10, itemized points. These conclusions cannot be drawn based on Figure 5. An increase in wind speed is not observed "at all times". It "may" be observed at any time, but certainly not at all times for all days. Similarly, cooling "may" occur between 1700 and 0000 UTC, but it certainly does not occur throughout that period for all cases. FLF-mean=1 does not occur "during the entire night" for all cases. If any of these statements were true, then we should see that the temporal occurrence equals the total number of days for a prolonged period of time in figure 5.

Page 11, line 3. "most probable" this means the time with the highest occurrence. Instead, the authors appear to refer to the median.

Page 11, line 5-6. "exhibited a nearly symmetric distribution centered at 1800 UTC." This distribution does not appear symmetric: it has a long tail towards later times. Also, it has a maximum at 1730 UTC.

Page 11, line 11. "clearly linked" – what is the reason for this statement? Visual inspection of the scatter plot? The authors should include a correlation value and its significance here.

Page 11, line 16. The "most frequent" onset seems to be at 1745 UTC, not 1900.

Page 11, line 22. "to reach 700" – this looks like 500 in Figure 8.

Page 11 line 23. "most likely" this again appears to be the "median", which is a different measure.

Page 11, line 31. Is there a reference for this "precious dataset"?

Page 12, line 14. "are said to occur" – please provide a reference.

Page 12, line 14. Should the G and B also be average values?

Page 12, line 33. "when they are large enough" – the clouds? Please specify.

Page 13, line 20. This section should be "LLC lifetime statistics". "Macrophysical characteristics" suggests a description of the thickness and liquid water path of these clouds.

Page 14, line 7. "LLC always form" this is not true. There are days where the NLLJ forms, but no LLC are observed. Perhaps: "on the days that LLC form, they appear more than 3 hours..."

Page 14, line 14-16. It appears as if the authors combined different days to produce their Figure 12. It would be a more reliable result if the authors could ensure that their sample is consistent across the three statistics, i.e. only select those days that have both a NLLJ and a cloud deck.

Page 14, line 23. "after sunrise or later" – either say "after sunrise" or be specific about "later", e.g. "after sunrise or up to X hours later".

Page 15, line 18. "the most frequent occurrence is at 1800 UTC" – this is true for the FLF-mean measure, but not the others. Is it really "most frequent" that is the useful statistic here? Why not report the median?

---

## Author Comment (AC1) · 27 Apr 2019

**Response to reviewer 1#**

**We thank the reviewer 1# for his/her valuable and constructive suggestions, which led to significant improvements of the quality of our manuscript. Below we detailed how his/her comments are addressed in the revised version of the manuscript. The corrections made in the manuscript and cited in this document appear in italic.**

**Review of Low Level Cloud and Dynamical Features within the Southern West African Monsoon by Cheikh Dione et al.**

**Summary of manuscript:**

The paper analyzes new observational data from a ground site at Savè, Benin, which was established as part of the DACCIWA campaign. The data analysis focuses on quantifying the diurnal cycle and intra-seasonal variability of factors related to the southern West African Monsoon, associated dynamical features like the maritime inflow (MI) and nocturnal low level jet (NLLJ), and the formation and breakup of low level stratiform clouds.

Monsoon flow was found to occur at some point on every day studied, with the strongest flow occurring at night. Onset of both the MI and NLLJ occurred most frequently between 1600-2100 UTC and breakup of the NLLJ occurred most frequently around sunrise. The distribution of MI arrival times was shifted earlier than expected considering the distance travelled, but strong monsoon flows may explain this result. The paper highlights the difficulty of cleanly separating MI and NLLJ phenomena in observations.

Low level stratiform clouds formed on 65% of the nights studied and usually broke up by 1200 UTC the following day as the planetary boundary layer became more turbulent and deepened. Cloud bases were typically formed near the core of the NLLJ.

The manuscript is well-organized but lacks sufficient detail and clarity in discussing its methods and reporting its results. The data analysis itself seems to be on solid footing but the manuscript requires a substantial amount of editing to provide a clearer accounting of the analysis and its significance.

In addition, a significant amount of further proofreading and editing is necessary for missing units, unlabeled elements on figures and tables, figure legibility, citation format, general typos, and grammar.

Recommendation: I recommend acceptance following adequate revision of the manuscript for clarity and completeness in reporting its methods and results.

**Major issues:**
1. It would be more accurate in your title and throughout the paper to refer to "low level stratiform clouds" or something similar rather than "low level clouds." The paper explicitly limits its analysis to stratiform boundary layer clouds and does not dwell on the shallow cumulus clouds that form after the stratus breaks up during the day. These low-level cumuliform clouds also may be worthy of future study for radiative and other implications.

We agree with reviewer #1 that the clouds under study are stratiform clouds. We corrected the title and the text as suggested.

2. Page 1, Line 15: The statement "Monsoon flow is observed 100% of the time" seems to contradict Figure 1 and the definitions in the paper. Did you mean that monsoon flow is observed at some point on 100% of days studied? Such a statement would be supported by the data provided.
The statement "monsoon flow is observed 100% of the time" is misleading. The monsoon flow is observed every day at Savè/Benin during our study period but not all the time. The sentence is now: *"Monsoon flow is observed every day during our study period"*.

3. Figures and tables, generally: There are major issues with clarity on several figures and tables.

a) Figure 2: The gray markers are extremely difficult to distinguish against the background shading. A more contrasting color, such as gold, could make the figure more legible.
We changed in the new version the gray markers to dark red. We hope this figure is now more legible.

b) Figure 4: The vertical lines are not labeled in the figure, described in the caption, mentioned in the manuscript. My impression is that the solid black line is for the FLFmean MI threshold, the grey dotted line for the potential temperature MI threshold, and the black dashed line for the NLLJ threshold, but this should be made explicit within the figure or in the caption. In addition, there is no indication of what the horizontal dashed line signifies, although it appears to be the zero marker on the Height axis and also a separator between the z-t plots and the FLF plots. As before, the grey markers are barely legible against the background shading.
We revised this figure as suggested and added more information in his label.

c) Figure 10: As before, the grey color is incredibly hard to discern. Here it's possible that thickening the lines would be sufficient, although choosing a different color that offered more contrast would work as well.
We have thickened the gray line, and used a darker gray for more clarity.

d) Table 1: It would be helpful to specify in the caption that the FLFmean criterion was used in the MI onset column. In addition, "DC" is never defined in the caption or the text, although I'm assuming it stands for "density current."
We thank the reviewer for this comment. We added the meaning of DC in the legend of Table 1, and the information about the criteria used for the fuzzy function

**Specific comments:**

1. Title: It would be helpful to add "Observed at Savè, Benin" at the end of the title to better describe the paper. It's not clear from the present title whether the study will focus on model results, satellite observations, site-specific observations, etc., and the most significant portion of the paper is the description of novel observations taken at the site.
We thank the reviewer for this suggestion which will clarify the objective of the paper. The new title is *"Low Level Stratiform Clouds and Dynamical Features observed within the Southern West African Monsoon"*. However, as reviewer #1 can see, we did not add the location of the observations because it

makes a too long title. This information is now clearly specified at the very beginning (Line 7) of the summary.

2. Page 1, Line 5: What does the term "quantitative documentation" mean in this context? Is it that the clouds are not well-simulated, or that not enough has been published about the simulated cloud properties?
We meant that quantitative and precised description of the low troposphere and clouds from the observations were missing before DACCIWA, for the models to be able to improve their ability to represent them properly.
We have revised the abstract accordingly:
*"Moreover, numerical climate and weather models need a finer description and knowledge of cloud macrophysical characteristics and of the dynamical and thermodynamical structures occupying the lowest troposphere, in order to be properly evaluated in this region"*

3. Page 1, Line 16: It's not clear what "According to monsoon flow conditions" means in this context. You mention the correlation with monsoon flow strength in the next line, which seems to make this phrase redundant.
Sorry, we meant "according to synoptic atmospheric condition". We corrected this in the new version.
We have reworded the sentence to avoid redundancy:
*"The maritime inflow reaches Savè around 1800-1900 UTC on average. This time occurrence is correlated with the strength of the monsoon flow"*

4. Page 1, Line 23: Perhaps "and intra-seasonal" should be added in between "day-to-day" and "variability" given the importance of the different monsoon phases and synoptic setups (e.g., vortex circulations).
We agree with the reviewer and we corrected the sentence as suggested.

5. Page 2, Line 13: It's confusing to distinguish between "aircraft" and "field" campaigns — aircraft campaigns are generally considered a subset of field campaign. For example, a NASA data archive defines an atmospheric field campaign as "an observational study planned for a specific location and a defined time period during which measurements are conducted from airborne platforms and/or ground sites to study physical and chemical processes in the atmosphere" (https://eosweb.larc.nasa.gov/field-campaigns). "Ground-based" may be a more appropriate phrase for the supersite data.
Thank you for the reference website for the definition of the field campaigns.. We reworded the sentence: *"Filling the gap of observations and studying the LLSC life cycle were therefore the primary goals of the Dynamics-Aerosol-Chemistry-Cloud Interactions in West Africa (DACCIWA) project (Knippertz et al., 2015) with an aircraft and ground-based campaigns (Flamant et al., 2017, Kalthoff et al., 2017) performed during summer 2016."*

6. Page 2, Line 13 (and throughout): The citation year for the Kalthoff et al. Paper should be 2018 instead of 2017 to refer to the published version.
Yes, this has been corrected.

7. Page 2, Line 16: Why are only the data from Savè used? It would be helpful to more fully motivate the decision to focus on this site in particular when two others are theoretically available as well.
The Savè super site was the most instrumented site. Sodars were implemented at Kumasi and Ile-Ife super sites but they provided wind only up to 300 m, preventing the study of the low level jet.

Furthermore, no cloud radar was available at these two sites, and the cloud summit could not be determined properly.
We added a sentence to explain this in the text:
 *"The focus on Savè for this study is motivated by the fact that only this site was instrumented in such way that continuous profiling of the wind up to several kilometers and continuous determination of the cloud summit were accessible. This was not the case at Kumasi or Ile Ife"*

8. Page 4, Line 1: Please define COSMO before using the acronym.
We defined COSMO in the revised version of the manuscript: *"COSMO (Consortium for Small-scale Modeling)"*

9. Page 4, Line 18: "On the one hand... on the other hand" generally signifies that two things will be contrasted, but that is not really the case in these sections. A re-write to "Section 4 presents results for the NLLJ and MI and Section 5 for the LLC" or something similar would be better.
We agree. The sentence is now as suggested: *"Sections 4 presents the results for the NLLJ and MI and section 5 for the LLC."*

10. Page 5, Line 14: It is not clear what "above and below" 1000/2000 m means in this sentence. Is it 0.5 K between 550-1000 m and 2 K between 1000-2000 m? Or something similar? Please clarify.
We agree that the sentence is unclear. . We clarified the sentence and reworded this paragraph in the revised version of the manuscript: *"A systematic comparison of the radiosounding temperature profiles with the HATPRO temperature profiles (not shown) revealed a systematic cold bias of 0.2 K below 550 m, 0.5 K in the 550 - 1000 m layer, and 2 K in the 1000 -2000 m layer."*

11. Page 6, Line 29: The vortex circulations, deep convection, and Harmattan flow are filtered out, or excluded, from the analysis, correct? Just saying "filtered" is ambiguous about whether these observed values are excluded or somehow corrected.
We reworded this sentence in the revised version as:
*"This criterion on wind direction allows to exclude atmospheric conditions associated with vortex circulations, deep convection and Harmattan flow."*

12. Page 7, Line 9: The phrase "found situations to be true" is missing some critical information. What did the simulations find to be true? In context it seems that the Couvreaux et al. paper is cited to support the previous assertion about linking synoptic setups to monsoon variability. Perhaps it would be better to just cite the paper at the end of that sentence if you're not making any further points about the study?
We followed this suggestion and cited Couvreux et al. (2010) at the end of the sentence."

13. Page 7, Line 34: It's a bit of stretch to say that Figure 3c indicates a "clear diurnal cycle" in wind direction. Can you in some way quantify that there's a statistically meaningful diurnal difference? It seems likely to me the difference is real, but it's not self-evidently true.
We thank the reviewer for this comment. Instead of certify that there is a "clear diurnal cycle" we quantify the turn in wind direction along the day with an amplitude around 32°. The paragraph is now: *"The median strength of the monsoon flow is roughly 3.5 m s$^{-1}$ between noon and 1700 UTC with a 210° direction. The median strength regularly increases between 1700 and 0100 UTC up to 5.5 m s$^{-1}$ with a simultaneous slight shift in the median wind*

*direction (amplitude around 30° and standard deviation around 9.46). These same changes are observed in wind surface measurements (Kalthoff et al., 2018)."*

14. Page 8, Line 18: It would be helpful to put "This last criterion" or something similar here to make clear it's only the third criterion that "ensures stable to neutral conditions at the surface."
We agree with the reviewer and modified the sentence accordingly:
*"This last criterion ensures stable to neutral conditions at the surface."*

15. Page 8, Line 30: When introducing the fuzzy logic method, it would be helpful to motivate why this method is necessary/helpful. From the rest of the paper it seems like the 302 K potential temperature threshold works just as well, so the main benefit is the ability to look at wind and temperature components separately?

It is true that 4 criteria are used to determine the MI arrival time and that could seem a bit confusing. The 302K isentrope was first used by Deetz et al.. This criteria has no clear and objective justification, except that it works well on one cases simulated by Deetz et al. We wished to propose a criterion built on MI characteristics as observed at surface in Savè and which are an increase of wind and a decrease of temperature. Fuzzy logic function seemed to be the most efficient way to study at the same time each change separately and merge their effect in one criterion. We did not wish to remove totally the 302K criterion since it already has been published and finally gives some quite consistent MI occurrence time.

We added this sentence to explain why we suggest a new criterion: *"Since the 302K criterion relies on one simulated case and because no clear justification for this 302K value exists, another criterion based on MI characteristics observed at surface is proposed in this study. MI arrival at Savè should be detected at surface by a combination of both an increase in horizontal wind and a decrease in temperature."*

16. Page 9, Line 11: It would be helpful to rewrite Equation (1) here plugging in the values for y1, y2, and ($r_x$):

$$FLF_x(r_x) = \begin{cases} 0, & r_x \leq 0 \\ \dfrac{r_x}{(r_x)_2}, & 0 < r_x < (r_x)_2 \\ 1, & r_x \geq (r_x)_2 \end{cases}$$

We changed the equation 1 as suggested by the reviewer. However, the numeric values of $r_{x2}$ depend on the time series of the temperature or wind speed modifications. For each day, $r_{x2}$ is the value rx corresponding to 99 percentile. We reworded this paragraph as: *"where $r_x$ is the rate of change of the variable x, $r_{x1}$ (resp. $r_{x2}$) is a constant value below (above) which $FLF_x$ is equal to y1b (y2 ). $r_T$ is multiplied by −1 to obtain positive changes for decreasing temperature. As in Coceal et al. (2018), y1 and y2 are set to 0 and 1, respectively and $r_{x1}$ is set to 0 (i.e., no increase in wind speed or no decrease in temperature). Instead of using the maximum value of $r_x$ divided by two for $r_{x2}$ (Coceal et al., 2018), for each day, we use the value corresponding to the 99-percentile of $r_x$ divided by two to avoid outliers."*

17. Page 9, Line 11: The definition for the mean FLF function should specify whether you're averaging the two other FLF functions (my impression) or taking rx as the average of the wind speed and negative temperature tendencies.

The mean FLF function is define as the mean algebraic average of the two other FLF functions. We rewrote this sentence as:

*"In this study, the mean fuzzy logic function $FLF_{mean}$ is computed using equal weights for $FLF_T$ and $FLF_{ws}$, and the same threshold of 1 is used to detect combined changes in the dynamic and thermodynamic conditions."*

18. Page 9, Line 13: I can't tell what this sentence about the fuzzy logic method being "meaningful" is actually saying. Meaningful in what sense? Is there some evidence that you want to highlight about this being a meaningful metric?

We intended to say that the use of a fuzzy logic function only on wind speed or temperature changes does not make sense: detection of temporal gradient would have been sufficient. We changed the sentence as follow:

*"The fuzzy logic method only makes sense if the temperature and wind speed changes are combined."*

19. Page 10, Line 8: Is this supposed to be criteria ii)? Also, as written on Page 8, criterion ii) does not make clear the maximum wind speed must be below 500 m, just that the maximum wind speed below 500 m must be at least 5 m/s. You should clarify this criterion.

*T*he misunderstanding comes from an unclear explanation of the criteria for the NLLJ detection. The NLLJ core below 500m is only used at the settlement of the NLLJ, since the core rises in height with time. We clarified the paragraph which details the criteria for the NLLJ detection (page 8): *"The detection of the NLLJ is based, in this study, on the use of dynamical and surface stability criteria: (i) the wind direction in the lowest atmosphere below 1500 m is between the south-east and west-northwest with (ii) a maximum wind speed of at least 5 m s−1 and at least 2 m s−1 larger than the minimum above and (iii) a surface sensible heat flux lower than 10 W m−2. This last criterion ensures stable to neutral conditions at the surface. The onset of the NLLJ is defined when these criteria are satisfied for at least two hours and the height of the maximum wind speed is below 500 m. The breakup time is defined when one of the three criteria mentioned above has not been satisfied for at least 1 hour. The use of the surface sensible heat flux as a diagnostic of the stability may be a limitation to this method because this measurement is very local and may not represent atmospheric stability on large spatial scales."*

20. Page 10, Line 14: It is not clear what "if the same scenario appears every day" means here. If every day had the same scenario, it seems like it would be quite easy to determine solid criteria. This sentence should either be written to more clearly state its point or be deleted.

We agree with the reviewer that this part of the sentence is not understandable. We decided to simply remove it. It gives:

*"Based on these three examples, one can note that large differences occur that make it difficult to determine solid criteria for the detection of the MI and NLLJ."*

21. Page 10, Line 21: There is a notable period of muted wind speed increases in the morning between 0600-1200 UTC. This seems to contradict the "all times of day" phrasing. In addition, there are also spikes above 5 just before 6 UTC that complicates a simple 1700-0000 callout.

We agree with the reviewer that 'all times of days" is not correct. We modified the sentence accordingly:

*"A large increase in wind speed (F LF$_{ws}$ = 1) is observed at several times during the day; the largest ($\geq$ 5) occurs between 1700 and 2000 UTC. This large range of times is due to the day-to-day variability of the monsoon strength and the arrival time of the NLLJ during this period."*

22. Page 11, Lines 3 and 23: The phrases "most probable" and "most likely" suggest some kind of statistical analysis, although none is carried out, or at least documented. If these are conclusions just from visual inspection of Figures 6 and 8, it would be better to say something more along the lines of "most observations fell between the values of...". If you have some threshold (interquartile range? two standard deviations?) being used to define "most probable" or "most likely," it should be reported.

We rewrote this paragraph in the revised version as *"The MI arrival times are shown in Fig. 6a. Four estimates of the MI arrival times are displayed, one using the 302 K potential temperature criterion and three corresponding to the first time when the three fuzzy logic functions, FLF$_{ws}$, FLF$_T$ and FLF$_{mean}$, attain a value of 1. Most observations of the MI arrival time at Savè, considering only the wind speed increase, fell between 1600 and 1800 UTC; it is between 1600 and 2100 UTC when we consider only the cooling. The arrival time of the MI deduced from FLF$_{mean}$, which couples with an equal weight cooling and wind speed increase, exhibited a strong variability. There are earlier arrival times at 1600 UTC and later ones at 0630 UTC. As noted above, the different tests performed to select the constants and thresholds for the fuzzy logic method yield different MI detection times for each day but quite similar distributions for the period of study. These results suggest that the MI arrival time is difficult to detect with local measurements. However, the MI arrival time as detected by the fuzzy logic function FLF$_{mean}$ is clearly linked to the mean monsoon flow in the afternoon (Fig. 7): the stronger the monsoon flow strength in the afternoon between 1200 and 1500 UTC, the earlier the MI arrival time. The two exceptionally early arrivals at 1600 UTC shown in Fig. 7 are associated with unusually strong monsoon flow all day long (e.g. the night 9-10 July illustrated in Fig. 4c)."*

23. Page 11, Line 24: It's not clear how you reach the conclusion that the NLLJ cores from AMMA would have higher wind speeds if they were the same height as those observed in this paper, or what the implications of this are.

We agree with the reviewer that this sentence was not clearly rewritten and we reworded it:

*"The NLLJ core in Niger and Benin observed during AMMA campaign (Lothon et al., 2008) was roughly at the same height, but the wind speed of the jet core is in contrast more important in Niger, from 10 to 20 m s$^{-1}$."*

24. Page 12, Line 14: Should G and B also be the average values in criterion i)? If not, what fraction of pixels must satisfy criterion i) for the scene to be considered cloudy?

G and B are indeed average value in (i) and we corrected it in the new version, with added over lines.

25. Page 13, Line 30: I can't find where earlier in the paper it's mentioned that LLCs cannot be determined during rain events. From Figure 10 it appears that the IR camera continued to collect valid data. Please clarify either here or in an earlier section.

We thank the reviewer for this comment. We missed to mention that LLCs cannot be determined during rain events with infrared camera. We added the following sentence in the new version of the manuscript to clarify this section:

*"Note that during rain events, droplets are retained on the dome of the infrared camera and impact the color of the image as if there was a cloud. Therefore, LLSC cannot be detected during rain events*

*which are thus excluded. As far as we know, it is the first time that such methodology is used for the study of the stratus cloud deck formation and breaking."*

26. Page 14, Line 17: What does "articulation" mean in this context? It's also unclear what exactly is "considered in the statistics."
We agree that the word "articulation" may not be appropriate here. We modified this sentence in the revised version as:
*"Figure 12 provides a schematic evolution of the NLLJ and of the low level stratus that we observe during the DACCIWA field experiment at the Savè site for the 25 days considered in the statistics."*

27. Page 14, Line 27: Did you mean to say "a key need for observations to compare with numerical weather and climate models" or something to that effect? The sentence is missing something as currently written.
Few words are actually missing and we apologize for this. We reworded the sentence:
*"A key need for observations to compare with numerical weather and climate models motivated the field campaign in the DACCIWA project."*

28. Page 15, Line 1: It would be helpful to discuss a bit how the MCSs impact the variability results in the paper versus simply asserting they're important, given this was not made much of a focus previously in the manuscript.

It is true that MCS were not our focus because their impacts are difficult to investigate with observations at one point. However, we tried to exclude some evident events when we detected density current or rain fall. We added a paragraph explaining what could be the impacts of MCS that could not be detected at Savè and then possibly included in the statistics analyzed in this paper.

*"MCS, when occurred in the surroundings of Savè site, could be detected (rain fall or density current) and excluded from the analyzed days. However, MCS occurring upstream of Savè were hard to detect and could have some more subtle impacts on the monsoon flow or the MI characteristics or propagation."*

29. Page 15, Line 8: Does the flow really turn to the south? From Figure 3, it looks like the flow becomes more southerly if anything, meaning the wind is turning to the north. Saying the winds become more southerly would be clearer.
We thank the reviewer for his suggestion. We did mean "southerly", and the text has been corrected.

30. Page 15, Line 29: It would be better to say that low level stratus clouds persist until noon on "80% of days with nocturnal stratus formation" or something similar. Otherwise it looks like this was observed on 80% of all days, which is problematic given that only 65% of days had nocturnal stratus cloud formation to begin with.
We thank the reviewer for his correction. These sentences are now:
*" Low level stratus is a persistent phenomenon that occurs 65% of nights. It forms more than 3 hours (6 hours in average) after the NLLJ onset at the jet core height and persists until noon on 80% of days with nocturnal stratus formation."*

31. Page 16, Line 4: This seems very abrupt and incomplete for a conclusion to the paper. The paper would be greatly improved with a final paragraph explaining the broader significance of this work and perhaps suggestions for future directions or uses for the data.

We agree with the reviewer and added the following sentences*: "The relative contribution of those processes is addressed in Adler at al. (2018) and Babic et al. (2018) for 15 IOPs and one case study respectively. Our work brings an overall statistical analysis of the key dynamical features of the low troposphere during the WAM. It also exhaustively gives quantified diagnostics for each day of the entire period. Therefore, it is an important basis for any future case study, and to model evaluation."*

32. Page 16, Line 5: You should state explicitly how long the DACCIWA embargo period will be. It would also be useful to provide a DOI or URL to the baobab database if available.

Today, there is no embargo anymore on the DACCIWA dataset. We added the URL to access to the baobab database. Dataset DOI are already listed in the reference:

*Derrien, S., Bezombes, Y., Bret, G., Gabella, O., Jarnot, C., Medina, P., Piques, E., Delon, C., Dione, C., Campistron, B., Durand, P., Jambert, C., Lohou, F., Lothon, M., Pacifico, F., Meyerfeld, Y.: DACCIWA field campaign, Savè super-site, UPS instrumentation; SEDOO OMP. https://doi.org/10.6096/DACCIWA.1618, 2016.*

*Kohler, M., Kalthoff, N., Seringer, J., and Kraut, S.: DACCIWA field campaign, Savè super-site, Surface measurements; SEDOO OMP. https://doi.org/10.6096/dacciwa.1690, 2016*

*Handwerker, J., Scheer, S., and Gamer, T.: DACCIWA field campaign, Savè super-site, Cloud and precipitation; SEDOO OMP. https://doi.org/10.6096/dacciwa.1686, 2016.*

**Technical corrections:** There are numerous issues of copy-editing (grammar, reference format, etc.) that need further review. Because of the importance of the abstract, I list all the issues that I identified here. I leave the remaining, similar errors to the authors to address in further proofreading unless the mistake impedes understanding or is in an important location (e.g., subhead).

1. Page 1, Line 2: "Boreal" should not be capitalized.
This has been corrected.

2. Page 1, Line 3: There should not be a comma after "land."
This has been corrected.

3. Page 1, Line 4: "These" should be used instead of "those."
This has been corrected.

4. Page 1, Line 9: "Continuous measurements collected" should be changed to "measurements continuously collected" or the "continuous" should be moved to before "in-situ" in the line above.
This has been corrected.

5. Page 1, Line 11: "Data" should not be capitalized.
This has been corrected.

6. Page 1, Line 20: "Stratus cloud" should be pluralized.
This has been corrected.

7. Page 3, Line 15: "Phase 3" should instead be "Phase 4."

This has been corrected.

8. Page 4, Line 28 & Page 5, Line 8: "Low troposphere" should be "lower troposphere."
This has been corrected.

9. Page 5, Line 14: There is a missing unit of "K" after 0.5.
This has been corrected.

10. Page 13, Line 27: Figure 10?
This has been corrected.

---

## Author Comment (AC2) · 27 Apr 2019

Review of the study "Low level cloud and dynamical features within the Southern West African Monsoon" by Dione et al.

General comment: This study aims at analyzing the dynamic and the variability of the nocturnal low level jet, the maritime inflow and their connections to the low level clouds thanks to high resolution wind profiler, observations and cloud monitoring. This study is correctly written and the results are clear and well presented. The only main comment is on the main objectives and the contribution of this study with respect to the DACCIWA project and the others studies/papers that should be better highlighted.
I recommend minor revisions with the list of detailed comments below.

We thank greatly the reviewer for his/her feedback. We have addressed them all below.

Abstract: I recommend to the authors to improve the abstract by removing some descriptions and better identifying and emphasing the main objectives and results of this specific study. The main contributions of this study to the DACCIWA project should be clearly mentioned. This is also true for the introduction.
We rewrote the abstract and introduction as suggested by the reviewer #2.

P2 l33, P3 l23, P3 l31 . . . Nested brackets.
This has been corrected.

Fig 1: Please clarify that the colors show the mean dew point temperature for all the period or only during the DACCIWA campaign. Also why the ITD is only displayed for June (P3l3).
The ITD is displayed for June because this month corresponds to the transition between pre-onset and onset phases of the monsoon flow. The objective was simply to show that the area of interest, Savè supersite in Benin, remains always in the southern side of the ITD, that is within the monsoon flow.
We improved the label of the figure in several ways and it is now: "Dew point temperature obtained from the ECMWF re-analyses at 2 m over Southern West Africa during June 2016, and (black solid line) mean intertropical discontinuity (ITD) position during that month. The ITD position is deduced from the dew point temperature isoline of 15° C. The stars indicate the location of the three supersites of DACCIWA ground campaign and, among them, Savè supersite where the dataset analyzed in this study has been acquired the period that the ITD mean are computed."

P5 l19 I am not used to using terms like "Ten Hz . . .", does it mean a high frequency measurement?

Ten Hz is actually a high frequency measurement for in situ measurements. We reworded the sentence to make it clearer: *"High Frequency measurements of air temperature, specific humidity, and three components of the wind were obtained (at 0.1 s time interval)."*

P6 l2: The justification of the increase of the cloud base definition is not clear. The reason mentioned is to also detect shallow convection during the morning. But this is not indicated in the objective of the study. Please clarify.

We agreed with you that this sentence is not clear. We rewrote it in the revised manuscript as *"However, a 1500 m height limit allows us to extend our detection of LLSCs during daytime when the stratus cloud base height rises due to the growing convective boundary layer."*

P7 l3 and Figure 2: this sentence is not clear. To avoid confusion, I recommend to change the color when the data are missing.

We thank the reviewer for his remark. We changed the color when the data are missing. Please refer to the new figure (Fig 2) in the revised manuscript.

P7 l27 and Fig. 3: Because there is a strong seasonal cycle during this period (according to Fig 2, it seems there is a thickening of the monsoon layer from the beginning of July, around the 10th), what is the results of a composite study when splitting before and after the 10th of July?

We agree with the reviewer #2 that such an analysis would have been interesting over the whole monsoon cycle to show pre-onset and post-onset periods. Big differences would then be expected. We think that the DACCIWA ground campaign is too short (one month and half) and mainly focused on the post-onset period to show the seasonal cycle with composite figures. Furthermore, the post-onset period includes some very peculiar sub-periods (wet westerly regime for example) which would strongly impact the composite figure because of the already limited days in the statistic.

Figure 4: Please clarify what the vertical lines represent.

We thank the reviewer for noticing the uncompleted label of figure 4. The label is now*: "Time-height sections of (color) wind speed and (arrows) direction from the UHF wind profiler, on the nights of (a) 2-3 July, (b) 7-8 July, and (c) 9-10 July 2016. The black open circles indicate the jet core height detected with a maximum wind speed of at least 5 m s$^{-1}$, the magenta rectangles indicate the height of the minimum wind speed above the jet core and pink open circles indicate the monsoon flow depth. The black, blue, and red lines indicate the three fuzzy logic functions of the wind speed, temperature and their mean, respectively. The dashed vertical lines indicate the MI arrival time with two different criteria: (1) FLFmean > 1 criteria (red), (2) 302K isentropy criteria (black). The vertical black line indicates the NLLJ onset. The horizontal dashed line stands for the zero km height."*

P11 l2: How do the authors explain the delay in between the FLF derived from the wind and the temperature?

It is not easy to answer this question. The MI arrival is naturally supposed to be manifested by both a change in wind and a change in temperature. However, by the time it gets to Savè, 180 km from the coast, it has possibly undergone several influences by the MCS activity, local processes, larger scale forcing,… so that the "front" is not so much of a "front", there is no strong gradient in temperature or wind, and they finally may both alter. We believe that this remains an open question.

Figure 6a: It is difficult to distinguish the black and brown bars, please change the colors.

We changed the colors and have now a new figure 6a.

P11 l9: Why not using the new ERA5 reanalysis with hourly resolution to detect and analyze briefly the large scale detection of the MI and to better understand these difficulties at local scale?

We had studied this aspect before with ECMWF re-analyses. But the gradients turned out to be very slack and not well marked. So that it was not an easier technique, although well appropriate due to its spatial point of view.

P11 l13 and Fig. 7: Could you add and discuss the impact of the dates by adding colors for each dot? The monsoon strength and the MI arrival time should be related to the seasonal cycle. Isn't it?

Since all the arrival times of the MI are given in table 1, it is easy for the reader to find the date and all the characteristics of the day. Therefore, we preferred not to load the figure with the dates. We did not find any trend of the MI arrival time (as can be seen in Figure 10) and monsoon strength along the DACCIWA campaign, certainly because of the "short" observation period and the two interruptions due to peculiar regimes (as discussed previously). However, we revised our discussion in this paragraph, also following Reviewer 1# comment: *"However, the MI arrival time as detected by the fuzzy logic function $FLF_{mean}$ is clearly linked to the mean monsoon flow in the afternoon (Fig.7): the stronger the monsoon flow strength in the afternoon between 1200 and 1500 UTC, the earlier the MI arrival time. The two exceptionally early arrivals at 1600 and 1630 UTC shown in Figure 7 and put in Table 1 are associated with unusually strong monsoon flow all day long (e.g. the nights 10-11 and 11-12 July)."*

P11 l21 and Fig. 8: same comment as previously. Is there any difference between June and end of July?

As explained in the response to previous comment, we think that the period is too short to notice a trend. The intra-seasonnal variation is mainly due to special regimes or MCS perturbations.

P11 l21: The authors do not discuss the intensity of the wind that is maximum at 2.00 am. I expected later as shown by Ruchith and Raj (2015). Ruchith, R. D., & Raj, P. E. (2015). Features of nocturnal low level jet (NLLJ) observed over a tropical Indian station using high resolution Doppler wind lidar. Journal of Atmospheric and Solar-Terrestrial Physics, 123, 113-123..

We agreed that the median of the wind speed of the jet presents a local maximum at around 02:00 UTC, but we didn't discussed it because the increase of the median is very small before this time.

P13 paragraph starting l27: The authors suggest the impacts of the difference phases of the monsoon. Why this is not taken into account this when calculating the composite studies?

We didn't take into account the intra-seasonal variability of the monsoon. This paper focuses on a statistical analysis of phenomena observed over the lowest troposphere of the southern part of West Africa during a specific period. The phases, we are talking about are: pre-onset with 3 days, post-onset with the largest period (23 days) but interrupted by the vortex phase (4 days) and the wet westerly regime (5 days) (figure 2). It seems difficult to make comparable composite figures with such different statistic representations.

P14 l29: The authors mention other sites in introduction but it seems they are not used in that study. So it is not necessary to mention them and to put them on the map Fig.1.

We agree and corrected the figure 1 accordingly.

---

## Author Comment (AC3) · 27 Apr 2019

Review of "Low-level cloud and dynamical features within the southern West African monsoon" by Cheikh Dione et al.
Review summary
The authors describe the occurrence statistics of the nocturnal low-level jet, the maritime inflow, and stratus deck during a 40-day observation period of the DACCIWA campaign at Savè, Benin. The jet and inflow are identified from UHF wind profiler measurements and radiometer-derived temperature profiles. The stratus deck is identified using IR "RGB" measurements. The authors identified the dynamical features on 20-25 days out of 40 (when there was no appreciable precipitation or density current) and the stratus deck on most of those days. Comparing the onset and breakup, the stratus deck was found to initiate approximately 3 hours after the onset of the maritime inflow, while the breakup of the jet was found to occur around sunrise and the breakup of the stratus after sunrise.

The manuscript is generally well-written with a clear structure. The figures present some nice and valuable results, although the figures are sometimes too busy to easily interpret. Some of the statistical findings are reported using vague or incorrect terminology. Overall, the scientific concerns are minor and this paper could be accepted after minor revisions.

Major comments
1. Conclusion / novelty
The authors lean a lot on the work presented in Adler et al. (2018) and Babic et al. (2018). When reading the conclusions, the emphasis appears to be on the findings of those two papers, e.g. the list starting at line 31, page 15. Instead, the authors should highlight in their conclusion how their Figure 12 synthesizes their results. For instance, one could identify three key periods, e.g. 1800-0000, 0000-0700, and 0700-1200 (more or less as done in lines 10-24, page 14). This Figure 12 provides a broader context for the case-study type and process analysis done in the previous studies. The main point here is that from the conclusions, it is unclear what the novelty is of this particular paper (although the introduction does provide this in line 13-19 on page 4).
We agree with the reviewer and reworded some parts of the introduction and conclusions.

2. Busy figures
The amount of information condensed into single figure panels is impressive, but it makes it very difficult to interpret some of these. One might imagine using these figures in a presentation and certain features will be difficult to highlight. Specific issues are:

Figure 2. The grey squares are difficult to see. How important are these for this figure? The rainy conditions could be presented in separate panels, although that would shrink

We agree with the reviewer the gray squares are difficult to see. However we think the information of rain fall is worth to be indicated for several reasons:
1/ rain fall prevents the detection of LLC with the infra-red camera,
2/ rain fall is sometimes an indication of perturbation of the monsoon flow or MI arrival.
That are the reasons why we decided to keep this information in Figure 2, but we have changed the color of the markers for rain to a more visible color. We hope this makes the figure 2 more legible.

Figure 2. Alternatively, the authors could present the rainfall information in a separate Figure using the same day-hour axes. If the information is not crucial to the paper, it could be provided as a supplementary figure. As it stands, the information is getting lost.
Please see the answer to the previous comment.

Figure 4. The three markers are difficult to discern in this figure. A solution could be to (1) remove the wind barbs to a separate figure and (2) replace the open markers with slightly larger, filled, black markers of different shapes. As a separate point, the barbs are not intuitive to interpret, as they are shown against a height axis. If the barbs were placed in a separate panel, the authors could also colour code them or use a filled contour plot to emphasize different cardinal or intercardinal directions.
We agree with the reviewer that the three markers are difficult to discern in Figure 4. We made a new version of this figure using readable markers.

Figure 10. Although the figure is visually fun, it is difficult to read. The preceding analysis means that the colours are no longer necessary. Without the colors, it would be much easier to interpret the relationship between the onset and breakup of the jets and clouds.
We have thickened the gray line, and used a darker gray for more clarity of this figure. This allows an easier interpretation of the relationship between the onset and the breakup of the jet and cloud.

Minor comments

Page 3, line 4. Please mention the source of "dew point temperature" used in this paper.
The dew point temperature are from ERA_interim data. We added in the new version the following sentence: " The ITD mean location in June 2016 is indicated in Fig. 1 and estimated using the 15 ◦ C dew point temperature from ERA interim reanalysis (Buckle, 1996)"

Page 3, line 18. Please rephrase or clarify in the text what is meant by "convective turbulence".
We rephrased this sentence as follows: "A second very important dynamical feature is the NLLJ, which typically forms over land at the end of the day when turbulence in the convective boundary layer has ceased."

Page 3, line 20. What are "intertial oscillations"?
We meant frictionless inertial oscillations. We corrected the sentence in the new version of the manuscript. The sentence is now: "However, due to the low latitude and the low Coriolis force in the DACCIWA region, frictionless inertial oscillations above the nocturnal inversion layer might not be applicable'.

Page 3, line 23. What does "it" refer to? The NLLJ?
The reviewer is right; "it" does refer to NLLJ. We replaced "it" by "The NLLJ".

Page 5, line 6. Please give the exact limits of the profiler data, rather than "roughly 150 m".

The first available wind measurement from the profiler is actually precisely at 150 m a.g.l. So we removed "roughly" in this sentence.

Page 5, line 14. "above and below" this phrasing does not make sense. It suggests that the bias is both 0.5 and 2.0 K between 1000m and 2000m.
We reworded this sentence as the following "*A systematic comparison of the radiosounding temperature profiles with the HATPRO temperature profiles (not shown) revealed a systematic cold bias of 0.2 K below 550 m, 0.5 K in the 550 - 1000 m layer, and 2 K between 1000 m and 2000 m.*

Page 5, line 15. "funding" should be "finding".
The text has been corrected.

Page 5, line 19-24. Given that the UHF data are block-averaged to 15 minutes, and given that it is interpreted alongside the sensible heat flux. Shouldn't the latter also be block-averaged to 15 minutes? Please specify the averaging performed on these data.
We thank the reviewer for his comment on the different block-averaging periods used for UHF and sensible heat flux. Fifteen minutes averaging period is not long enough to include the large eddy contribution to the turbulent flux with a sufficient statistic. Therefore, 30 minutes sample has been chosen for this study. That means that the temporal resolution for NLLJ arrival time is 30 minutes. We added the following sentence at the end of the paragraph about NLLJ detection: "The NLLJ arrival and breakup times are determined with a 30-min temporal resolution, which corresponds to the sample duration for the sensible heat flux estimation."

Page 5, line 30. "manufactured" should be "manufacturer"
The text has been corrected.

Page 6, line 7. "most of the time" and "complementary scans". Please be specific. Did the radar perform a volume scan every 30 minutes? How long did the volume scan take, e.g. 5 minutes? Does that mean that you have five 5-minute estimates of cloud-top height per 30-minute period?
We rephrased this sentence to clarify the acquisition mode: *"It was run with vertical pointing every 5 minutes and horizontal scans every 30 minutes."*

Page 6, line 14-22. This analysis seems really nice and original. Is it designed in this study? Perhaps the authors could emphasize this more. If not, please provide references.
Yes, this analysis was designed in this study for the first time, so there is no reference to cite for it. We emphasized on this as follows: "This instrument is used here to study the horizontal homogeneity of the cloud deck and to define the onset and breakup times of the stratus deck, with a newly designed method."

We also added a sentence at the end of the description of the method: "As far as we know, it is the first time that such methodology is used for the study of stratus cloud deck formation and breaking."

And we finally emphasized on this aspect in the conclusion.

Page 6, line 29. Should "height" be "top" (of the monsoon flow).
We meant "top". This has been corrected.

Page 6, line 33. "depth" should be "layer".
We meant "height".

Page 7, line 29. The monsoon depth is less than 1500m in the middle of the night.
We changed the text as follows: " The median of the monsoon depth shows a weak diurnal evolution from a minimum value of 1200 m a.g.l. during the night to 2000 m a.g.l. during convective conditions (Fig. 3a), with a day-to-day variability"

Page 7, line 31-33. The authors use a reference from 2010 to describe the status of the monsoon in their 2016. Please consider rephrasing this sentence.
We agree that the sentence was misleading. It was a comment on what was observed during AMMA in early monsoon season. However, this comment is not essential and has been removed.

Page 8, line 18. It is important here to note the temporal resolution of surface sensible heat flux, if it is different to the other measurements (see previous comment for page 5).
As explained in the response to a previous comment, the flux are estimated over 30 minutes because it allows a correct statistic of the large turbulent eddies which contribute to the vertical transfer of buoyancy. Doing this, we agree that the UHF and the sensible heat flux are not at the same temporal resolution. Consequently the arrival time of the NLLJ is determined with the coarser temporal resolution which is the sensible heat flux one, meaning 30 minutes.  We added a comment on the text about this: "The NLLJ arrival and breakup times are determined with a 30-min temporal resolution, which corresponds to the sample duration for the sensible heat flux estimation."

Page 8, line 29. How is the 302 K potential temperature measured? Is it based on the radiometer profiler? Please specify.
The 302 K potential temperature is actually based on the radiometer measurements. We have specified this more clearly in the revised version: "This criterion was applied to the temperature measured locally by the microwave radiometer at the Savè site in order to detect the arrival of the MI."

Page 9, line 15-16. "affected" should be "applied".
This has been corrected.

Page 9, line 18-20. Please enlighten the reader to what range of thresholds are appropriate for r-ws and r-T, and which values were chosen for the subsequent analysis.
We thank the reviewer for this comment. The paragraph presenting $r_{ws}$ and $r_T$ was actually incomplete. The numeric values of $r_{x2}$ depend on the time series of the temperature or wind speed modifications. For each day, $r_{x2}$ is the value rx corresponding to  99 percentile. We reworded this paragraph as: *"where $r_x$ is the rate of change of the variable x, $r_{x1}$ (resp. $r_{x2}$) is a constant value below (above) which $FLF_x$ is equal to y1 (y2). $r_T$ is multiplied by −1 to obtain positive changes for decreasing temperature. As in Coceal et al. (2018), y1 and y2 are set to 0 and 1, respectively and $r_{x1}$ is set to 0 (i.e., no increase in wind speed or no decrease in temperature). Instead of using the maximum value of $r_x$ divided by two for $r_{x2}$ (Coceal et al., 2018), for each day, we use the value corresponding to the 99-percentile of $r_x$ divided by two to avoid outliers."*

Page 10, line 1-2. "the wind maximum increases" – it is the "height" of the wind maximum that increases. Please rephrase.
Yes, we indeed meant that 'the height of the wind speed maximum' increased. This is reworded in the revised version.

Page 10, line 14. "if the same scenario appears every day" – surely, the authors mean that it is difficult to determine criteria if "different" scenarios appear each day? (i.e. the opposite)
*This paragraph was not clear enough. It is reworded in the revised version as "Based on these three examples, one can note the large variability that can be observed from one day to the other, which makes it challenging to define solid common criteria for MI and NLLJ detection."*

Page 10, itemized points. These conclusions cannot be drawn based on Figure 5. An increase in wind speed is not observed "at all times". It "may" be observed at any time, but certainly not at all times for all days. Similarly, cooling "may" occur between 1700 and 0000 UTC, but it certainly does not occur throughout that period for all cases. FLF_mean=1 does not occur "during the entire night" for all cases. If any of these statements were true, then we should see that the temporal occurrence equals the total number of days for a prolonged period of time in figure 5.
We thank the reviewer for his comment. We reworded this section in the revised version as
  - *A large increase in wind speed ($FLF_{ws}$ = 1) may be observed at any time during the day; however the largest occurrence (> 5) is between 1700 and 2000 UTC. This variability is due to the day-to-day variability of the monsoon strength and the arrival time of the NLLJ during this period.*
  - *As expected, cooling may occur between 1800 and 0030 UTC the following day. Contrary to the wind speed, whose fuzzy logic function reaches 1 but rarely remains at that value for several hours during the night, while the temperature fuzzy logic function reaches this value many times during the night. This trend implies continuous cooling (Fig. 4). This result is in accordance with the continuous decrease in temperature within the MI from north to the south discussed by Adler et al. (2018).*

Page 11, line 3. "most probable" this means the time with the highest occurrence. Instead, the authors appear to refer to the median.
We reworded this sentence in the revised version as "*Most of the observations of  MI arrival time at Savè considering only the wind speed increase fell between 1600 and 1800 UTC; while they fell between 1600 and 2100 UTC when we consider only the cooling.*'

Page 11, line 5-6. "exhibited a nearly symmetric distribution centered at 1800 UTC."
This distribution does not appear symmetric: it has a long tail towards later times. Also, it has a maximum at 1730 UTC.
We rewrote the paragraph in the new version of the manuscript: "The arrival time deduced from FLFmean, which couples with an equal weight cooling and wind speed increase, is observed between 1600 UTC and 0000 UTC, with nevertheless the most probable arrival time at 1800 and 2000 UTC."

Page 11, line 11. "clearly linked" – what is the reason for this statement? Visual inspection of the scatter plot? The authors should include a correlation value and its significance here.
We based on the correlation coefficient between the strength of the Monsoon and the arrival time of the MI. We added in the revised version the following sentence to clarify "The absolute value of the correlation coefficient between both is 0.61."
Page 11, line 16. The "most frequent" onset seems to be at 1745 UTC, not 1900.
This has been corrected. The most frequent onset is actually 1730 UTC.

Page 11, line 22. "to reach 700" – this looks like 500 in Figure 8.

*That is true, and this is a mistake. We corrected in the revised version as 'to reach about 500 m at NLLJ breakup time'*

Page 11 line 23. "most likely" this again appears to be the "median", which is a different measure.

We reworded this in the revised version; we changed "the most likely strength" by the "median strength".

Page 11, line 31. Is there a reference for this "precious dataset"?

As far as we know there is no specific reference for this dataset. However, some of these data are available on the DACCIWA (baobab) database.

Page 12, line 14. "are said to occur" – please provide a reference.

There is no reference for this. This is a clumsy wording for our own statement here. We reworded this sentence as *"Here, we set the occurrence of the low stratus clouds when..."*

Page 12, line 14. Should the G and B also be average values?

This has been corrected.

Page 12, line 33. "when they are large enough" – the clouds? Please specify.

We thank the reviewer for pointing out this unclear sentence. It is now splitted in two sentences as follows: "After 10:30 UTC, the cloud base rises and the fractioning of the cloud base (less steady red-pink color, large σRGB for a long duration of time) increases with the development of the convective boundary layer. It defines the end of the stratus LLC"

Page 13, line 20. This section should be "LLC lifetime statistics". "Macrophysical characteristics" suggests a description of the thickness and liquid water path of these clouds.

We thank the reviewer for this suggestion. The title has been changed.

Page 14, line 7. "LLC always form" this is not true. There are days where the NLLJ forms, but no LLC are observed. Perhaps: "on the days that LLC form, they appear more than 3 hours."

The use of the word 'always' is not appropriated, we removed it in the new version, and followed the reviewer suggestion. The sentence is now: "On the days that LLC form, they appear usually more than 3 hours after the onset of the NLLJ and clear up after the NLLJ breakup time (Figure 10)."

Page 14, line 14-16. It appears as if the authors combined different days to produce their Figure 12. It would be a more reliable result if the authors could ensure that their sample is consistent across the three statistics, i.e. only select those days that have both a NLLJ and a cloud deck.

Several approaches could be followed here, for an increase of statistics in each phase in one hand, or more consistency all along the diurnal cycle in the other hand. We have verified that the result was not altered if we considered only the days that have combined NLLJ and cloud deck.

[Figure]

*Figure1: as in figure 12 but for only cases that we combined NLLJ and cloud deck*

Page 14, line 23. "after sunrise or later" – either say "after sunrise" or be specific about "later", e.g. "after sunrise or up to X hours later".
The sentence is corrected in the new version:" It sharply increases after 0800 UTC when the convective boundary layer develops."

Page 15, line 18. "the most frequent occurrence is at 1800 UTC" – this is true for the FLF-mean measure, but not the others. Is it really "most frequent" that is the useful statistic here? Why not report the median?
We corrected by adding the median of the arrival time of the MI in the revised version. "
*The MI arrival time at the Savè site occurs between 1600 and 2100 UTC; the median occurrence time is at 1900 UTC."*

---

## Author Response (AR2)

We thank the three reviewers for their valuable and constructive suggestions, which led to significant improvements of the quality of our manuscript. Below we detailed how their comments are addressed in the revised version of the manuscript. The corrections made in the manuscript and cited in this document appear in italic.

**Reviewer #1**

Page 1, Line 3: Delete "Moreover."

This has been corrected

Page 1, Line 24: "Main" not "mains."

This has been corrected

Page 10: I still think it would be useful to have an equation 2 with the values plugged into equation 1 — it's really hard to intuitively see what's happening in equation 1 but the formula is quite simple and easy to understand once the ones/zeros are plugged in.

We thank the reviewer for his suggestion to add an equation 2 with the values plugged into equation 1. The simplified equation has been added in the new version.

**Report#2**

**Reviewer #3**

Review for "Low level stratiform clouds and dynamical features observed within the Southern West African Monsoon" by Cheikh Dione et al.

The authors have engaged very well with the reviewers' comments and the manuscript is now in a state that is nearly ready for publication. Two figures should still be adjusted to improve legibility, but these are minor (technical) corrections.

Figure 4:

Considering this figure in the context of the manuscript, the barbs are not essential to this figure. It appears that wind direction is relevant for the identification of the NLLJ, as described on page 9 lines 17-18, but this is not referred to again when discussing figure 4. The barbs therefore clutter the figure while their information is summarized by the black vertical line for NLLJ onset.

The panels also contain an unnecessary dashed line at zero height, to separate the coloured figure from the line plot below. I propose the following improvements:

1. Remove the wind barbs. If the authors believe this is essential information for the paper (which is possible, given the barbs inform the NLLJ detection), then they can be included as a separate, supplementary figure. To reiterate, the authors do not mention wind direction in their discussion of Figure 4, so it does not appear essential for this figure.

2. Slightly detach the coloured figure from the line plot below, so that the need for the dashed line is removed. It will allow the y-axis label (Height) to be centered on the coloured figure and allow a separate y-axis (FLF?) for the line graph below.

We thank the reviewer for his suggestion that can help us to improve the readability of Figure 4. We overplotted the wind direction in the figure to show the day-to-day variability of the monsoon flow characteristics and the NLLJ. The wind direction allows to understand the change in wind direction from westerlies to the easterlies above the monsoon flow, and then the minimum wind detected above the NLLJ. However, we agree with the reviewer that the information from the wind barbs isn't described anywhere in the document. Thus we added some comments in the text about the direction especially on the easterly wind. We updated the lower panels of the figure 4 following the suggestions.

Figure 10:

This figure does not require "the mean color of the IR cloud sky camera image" to be included. It is clear from Figure 9 how LLC onset and break are determined from the IR imagery, so the key point of Figure 10 is to show how LLC onset and break vary with NLLJ onset and break. The colours distract from this information.

Also, the black stars are difficult to discern, partly because of the colours, but also because markers are used for the four lines. Please remove the markers (currently squares) for those lines, so that the black starts stand out as the only markers of interest.

We agree with the reviewer that the key point of Figure 10 is showed on figure 9 concerning the LLC

onset and break up time  and on the fact that the colors distract from this information. Thus, we decided to modify this Figure following the suggestions of the reviewer. However, for the publication of our manuscript, we prefer the old version of this figure with mean color of the IR cloud sky camera.

Minor comments:

Page 4, line 1-2: Please include the Dee et al. (2011) reference for ERA-Interim:

Dee, D.P., Uppala, S.M., Simmons, A.J., Berrisford, P., Poli, P., Kobayashi, S., Andrae, U., Balmaseda, M.A., Balsamo, G., Bauer, D.P. and Bechtold, P., 2011. The ERA-Interim reanalysis: Configuration and performance of the data assimilation system. Quarterly Journal of the royal meteorological society, 137(656), pp.553-597.

We thank the reviewer for his suggestion. We added the reference in the new version of the manuscript.

Page 8, line 1: "3th" Should this be "1.5 UHF wind profiler gates"?

Here, we means "3rd UHF wind profiler gates".  The UHF wind profiler gates in his low mode are spaced by 75 m (i.e., his vertical resolution). The 3rd gate of this radar correspond with the height 225 m. We corrected in the new version of the manuscript.

Page 9, line 21: "satisfied for at least two hours" – this part of the NLLJ onset criterion is not reflected in Figure 4 and the discussion, e.g. page 11, line 3-4: "the wind vertical profile reaches the threshold of 5 m/s at 2100 UTC, which is the onset time of the NLLJ". Should the onset time be 2300 UTC, or does the two-hour criterion only apply to wind direction and/or surface sensible heat flux?

We agree with the reviewer and apologize for uncorrected legend of the figure 4. The label of the vertical lines indicating NLLJ onset and MI arrival time were switched.  We corrected  the caption of Figure 4 in the new version of the manuscript. *"Time-height sections of (color) wind speed and (arrows) direction from the UHF wind profiler, on the nights of (a) 2-3 July, (b) 7-8 July, and (c) 9-10 July 2016. A leftward horizontal arrow indicates an easterly wind, an arrow from bottom to top, a southerly wind. The black open circles indicate the jet core height detected with a maximum wind speed of at least 5 m s−1, the magenta rectangles indicate the height of the minimum wind speed above the jet core and red open circles indicate the monsoon flow depth. The black, blue, and red lines in the lower box indicate the three fuzzy logic functions of the wind speed, temperature and their mean, respectively. The vertical red and black dashed lines indicate the MI arrival time estimated using two different criteria: (1) the F $LF_{mean}$ > 1 criterion, and (2) the 302 K isentrope criterion, respectively. The vertical black line indicates the NLLJ onset."*

Dear Editor,

Please find below some modifications, we made on our manuscript for it improvement. All the modifications appear in blue in the document.

1. pg. 5, l 10: we removed "those structures;"

2. pg. 5, l 23: We replaced "studies" with *"data sets"*

3. pg. 5, l 12: We replaced "Fifteen" with "15"

4. pg. 6, l 13: We added the missing reference in ()?

5. pg. 6, l 20: We replaced "Frequency" with *"frequency"*

6. pg. 12, l 32: We replaced "important" with *"pronounced"*

7. pg. 15, l 17-18: We found that their is a repetition of lines 11-12, e.g. it is doubling, so we reworded the sentence L17-18 as: *"LLSCs always clear up after the NLLJ breakup time."*

8. pg. 15, l 31-32: We replaced "15" with *"11"*

9. pg. 16, l 10: We deleted "their"

10. pg. 16, l 8-9. We removed the end of this sentence "because rain and/or density currents were measured at Savè"

11. pg. 16, l 16: We replaced "stoppage" with *"break"*

12. pg. 16, l 16-18: We added at the end of this sentence a new reference *(Babic et al., 2019b)*

13. pg. 16, l 19.  We added *"depth"* after 1500-m, i.e. 1500 m.

14. pg. 17, l 2: We reworded this sentence as: *"It also depends on the capability of the turbulent mixing to balance  monsoon flow, which slows the front progression inland."*

15. pg. 17, l 3-4: We broke this sentence in two and reworded the last one as: *"They are sometimes difficult to distinguish."*